# Trichoplein binds PCM1 and controls endothelial cell function by regulating autophagy

Andrea Martello[1], Angela Lauriola[2], David Mellis[1], Elisa Parish[1], John C Dawson[3], Lisa Imrie[4], Martina Vidmar[1], Noor Gammoh[3] (iD), Tijana Mitić[1], Mairi Brittan[1], Nicholas L Mills[1,5], Neil O Carragher[3], Domenico D'Arca[2,*] (iD) & Andrea Caporali[1,**] (iD)

## Abstract

Autophagy is an essential cellular quality control process that has emerged as a critical one for vascular homeostasis. Here, we show that trichoplein (TCHP) links autophagy with endothelial cell (EC) function. TCHP localizes to centriolar satellites, where it binds and stabilizes PCM1. Loss of TCHP leads to delocalization and proteasome-dependent degradation of PCM1, further resulting in degradation of PCM1's binding partner GABARAP. Autophagic flux under basal conditions is impaired in THCP-depleted ECs, and SQSTM1/p62 (p62) accumulates. We further show that TCHP promotes autophagosome maturation and efficient clearance of p62 within lysosomes, without affecting their degradative capacity. Reduced TCHP and high p62 levels are detected in primary ECs from patients with coronary artery disease. This phenotype correlates with impaired EC function and can be ameliorated by NF-κB inhibition. Moreover, Tchp knock-out mice accumulate of p62 in the heart and cardiac vessels correlating with reduced cardiac vascularization. Taken together, our data reveal that TCHP regulates endothelial cell function via an autophagy-mediated mechanism.

**Keywords** autophagy; centriolar satellites; endothelial cells; GABARAP; SQSTM1/p62
**Subject Categories** Autophagy & Cell Death; Molecular Biology of Disease

## Introduction

Autophagy is an essential quality control function for the cell to maintain its homeostasis, through selectively degrading harmful protein aggregates or damaged organelles. Moreover, autophagy is a vital intracellular process for recycling nutrients and generating energy for maintenance of cell viability in most tissues and adverse conditions [1]. Basal autophagy mediates proper cardiovascular function [2]. Variety of cardiovascular risk factors can cause defective autophagy in vascular cells, producing high levels of metabolic stress and impairing the functionality of endothelial cells (ECs) [3]. Autophagy has been shown to regulate angiogenic activity and the release of von Willebrand factor from ECs [4]. Also, endothelial-specific deficiency of autophagy is pro-inflammatory and pro-senescent, as it promoted atherogenic phenotype in a murine model of atherosclerosis [5].

Specific autophagic receptors are responsible for selective autophagy by tethering cargo to the site of autophagosomal engulfment [6]. The recognition of ubiquitinated substrates is provided by molecular adaptors including p62/SQSTM1 (p62), which bind on one side to ubiquitin and, on the other end, to autophagosome-specific proteins (like members of the LC3/GABARAP/Gate16 family). The interaction between p62 and LC3/GABARAP bridges the autophagic machinery with its cargo, thereby fostering the selective engulfment by the autophagosome [7]. In mammalian cells, six ATG8 orthologues exist that are divided into the LC3 and GABARAP subfamilies which have a non-redundant function during autophagosome biogenesis. Specifically, LC3 subfamily members promote elongation of phagophore membranes, whereas GABARAP is critical in the closure of the phagophore membrane [8], and fusion of autophagosomes with lysosomes [9]. Recent studies demonstrated that a pool of GABARAP exists in the centrosome and peri-centrosomal region and regulates autophagosome formation during amino acid starvation [10].

The levels of p62 are regulated transcriptionally and through continuous degradation during basal autophagy. The defective autophagy, however, induces accumulation of p62, followed by the formation of aggregates [11]. Accumulation of p62 is further observed in human ECs in the cerebral cavernous malformation disease [12] and in human smooth muscle cells whereby p62 accumulation accelerated the development of stress-induced premature senescence [13]. Besides its role in autophagy, p62 is a scaffolding hub for the cellular signalling pathways involving NF-

1   University/BHF Centre for Cardiovascular Science, QMRI, University of Edinburgh, Edinburgh, UK
2   Department of Biomedical, Metabolic and Neural Sciences, University of Modena & Reggio Emilia, Modena, Italy
3   Cancer Research UK Edinburgh Centre, Institute of Genetics and Molecular Medicine, University of Edinburgh, Edinburgh, UK
4   Centre for Synthetic and Systems Biology (SynthSys), University of Edinburgh, Edinburgh, UK
5   Usher Institute, University of Edinburgh, Edinburgh, UK
    *Corresponding author. Tel: +39 059 2055610; E-mail: domenico.darca@unimore.it
    **Corresponding author. Tel: +44 131 2426760; E-mail: a.caporali@ed.ac.uk

kB activation, nerve growth factor signalling and caspase activation [14].

Trichoplein (TCHP) is cytosolic ubiquitously expressed 62 kDa protein identified as a keratin filament binding protein [15]. So far, the function of TCHP is deemed dependent on its partner proteins and their cellular localization. In proliferating cells, TCHP serves as a scaffold protein not only for appendage-associated Ninein, involved in microtubule anchoring at the mother centriole [16], but also for the centriole-associated Aurora kinase A activity, implicated in the destabilization of cilia [17]. Alternatively, in differentiated, non-dividing epithelial cells, TCHP translocates from the centrioles to keratin filaments, and desmosomes [15]. TCHP was reported to reside on the outer mitochondrial membrane (OMM), where it binds mitofusin2 (Mfn2), regulating the ER–mitochondria tethering and promoting mitochondria fission [18,19]. Moreover, increased levels of TCHP enable decorin evoked mitophagy [20].

It is still unknown what role TCHP plays in EC function, particularly regarding its localization and mechanisms of action. We report here that TCHP localizes in centriolar satellites and it has an unexpected role in controlling autophagy in ECs.

## Results and Discussion

### Lack of TCHP impairs endothelial cell function

Accumulating evidence links the intact autophagic responses with the preservation of cardiovascular homeostasis in several physiological and pathological settings [21].

To examine the impact of TCHP on endothelial function, we performed a Matrigel tubule formation assay. TCHP down-regulation (Fig EV1A) severely affected the tubule forming capacity of HUVECs *in vitro* (Fig EV1B) and the formation of vessels *in vivo* using Matrigel plugs (Fig EV1C). In agreement, a reduced level of TCHP also affects ECs migration as measured in the wound healing (Fig EV1D). To further dissect the phenotype of ECs lacking TCHP, we analysed the expression of a subset of genes controlling angiogenesis, inflammation and cell cycle. TCHP knock-down cells showed an increase of IL1β, IL6, IL8, MCP1, CDKN2A/p16 and CDNKNB/p14 expression (Fig EV1E) and displayed a senescent-associated phenotype as seen by the increase of CDKN2A/p16 (Fig EV1F), β-galactosidase activity (β-Gal) (Fig EV1G) and the accumulation of aggresomes at 7 days postlentiviral transduction (Fig EV1H).

### TCHP binds PCM1 to regulate its localization

To identify TCHP interacting partners, we performed co-immunoprecipitation (Co-IP) coupled with mass spectrometry analysis using FLAG-tagged TCHP as bait in HEK293 cells. The centriolar satellite protein PCM1 was identified as the most enriched protein in anti-FLAG pull-down in comparison with the control experiment (Table EV1 and Appendix Fig S1A). We validated the mass spectrometry results showing that in HEK293 cells, two different FLAG-tagged versions (N- and C- terminal) of TCHP co-immunoprecipitated with endogenous PCM1 (Fig 1A). TCHP displays two coiled-coil regions, at the N terminus (1–136 AA), which are necessary and sufficient for centriolar localization and function [22]. Using FLAG-tagged TCHP deletion mutants (Appendix Fig S1B), we found that the residues corresponding to the second coiled-coil region (41–136 AA) are critical for the binding to PCM1 (Fig 1B). Moreover, we demonstrated that the interaction between TCHP and PCM1 is conserved in HUVECs since TCHP is co-immunoprecipitated with endogenous PCM1 (Fig 1C).

Endogenous or expressed TCHP showed a dynamic localization in cells that could be due to a different repositioning of TCHP during different cell cycle stages or under the effect of cellular stressor or stimuli [15,17]. Moreover, THCP localization in different subcellular compartments may mirror different functional roles played by the same protein. We next tested the localization of PCM1 and TCHP in ECs. PCM1 was the first satellite protein identified [23] and is acting as satellite assembly scaffold for other centriolar satellite proteins such as Cep290 [24] and Cep72 [25].

We established that TCHP extensively co-localized with PCM1 in the pericentriolar matrix and satellite region (Fig 1D) and localized close to the nucleus in the same compartment occupied by the centriolar satellite proteins CEP290 and CEP72 (Fig 1D). Interestingly, depletion of TCHP had a significant effect on PCM1 localization, showing loss of PCM1 accumulation at the perinuclear region and dispersion throughout the cytoplasm (Fig 1E).

Overall, these data demonstrated for the first time that TCHP binds PCM1 and localizes in centriolar satellites. During the revision of this manuscript, an independent study was published reporting the spatial and proteomic profiling of 22 human satellite proteins using proximity-dependent biotin identification, including PCM1, CEP290 and CEP72 [26]. Consistently with these results, this unbiased approach has identified TCHP as part of the centriolar satellite protein network, directly interacting with PCM1, CEP290 and CEP72.

---

**Figure 1. TCHP interacts directly with PCM1.**

A, B  (A) Immunoblot analysis showed HEK293 transfected for 48 h with expression vectors for TCHP, and FLAG-TCHP at N- and C-terminal. (B) Immunoblot analysis showed HEK293 transfected for 48 h with expression vectors for TCHP-FLAG C-terminal or constructs with deletions of the coiled-coil domain 1 (TCHP Δ1) and 2 (TCHP Δ2), as indicated in the scheme. For (A and B), total lysates were immunoprecipitated with anti-FLAG antibodies, and blots were probed sequentially with anti-PCM1, anti-FLAG and anti-ACTIN antibodies. IgG light chains are indicated with Red Ponceau staining. The input totals were analysed by parallel immunoblotting as a control for the level of expression.

C  Anti-PCM1 immunoprecipitation from HUVECs cells and TCHP and PCM1 immunoblot.

D  Co-localization of TCHP-V5 and centriolar satellite proteins. HUVECs were transduced with TCHP-V5 lentivirus and stained for anti-PCM1, anti-CEP290, anti-CEP72 and anti-V5 antibodies. Scale bars, 25 and 5 μm in the inset. *Right panel:* the two-channel intensity correlation of pixels corresponding to regions identified with TCHP-V5 and satellite markers (n = 90 cells; Pearson co-localization coefficient).

E  TCHP knock-down or control cells were stained for anti-PCM1 antibody. *Panel below:* quantification (n = 50 cells; unpaired t-test; **P = 0.004 vs. control). Scale bars, 50 and 5 μm in the inset.

Data information: Statistical analyses were performed on at least three independent experiments. Data are represented as mean ± SD.

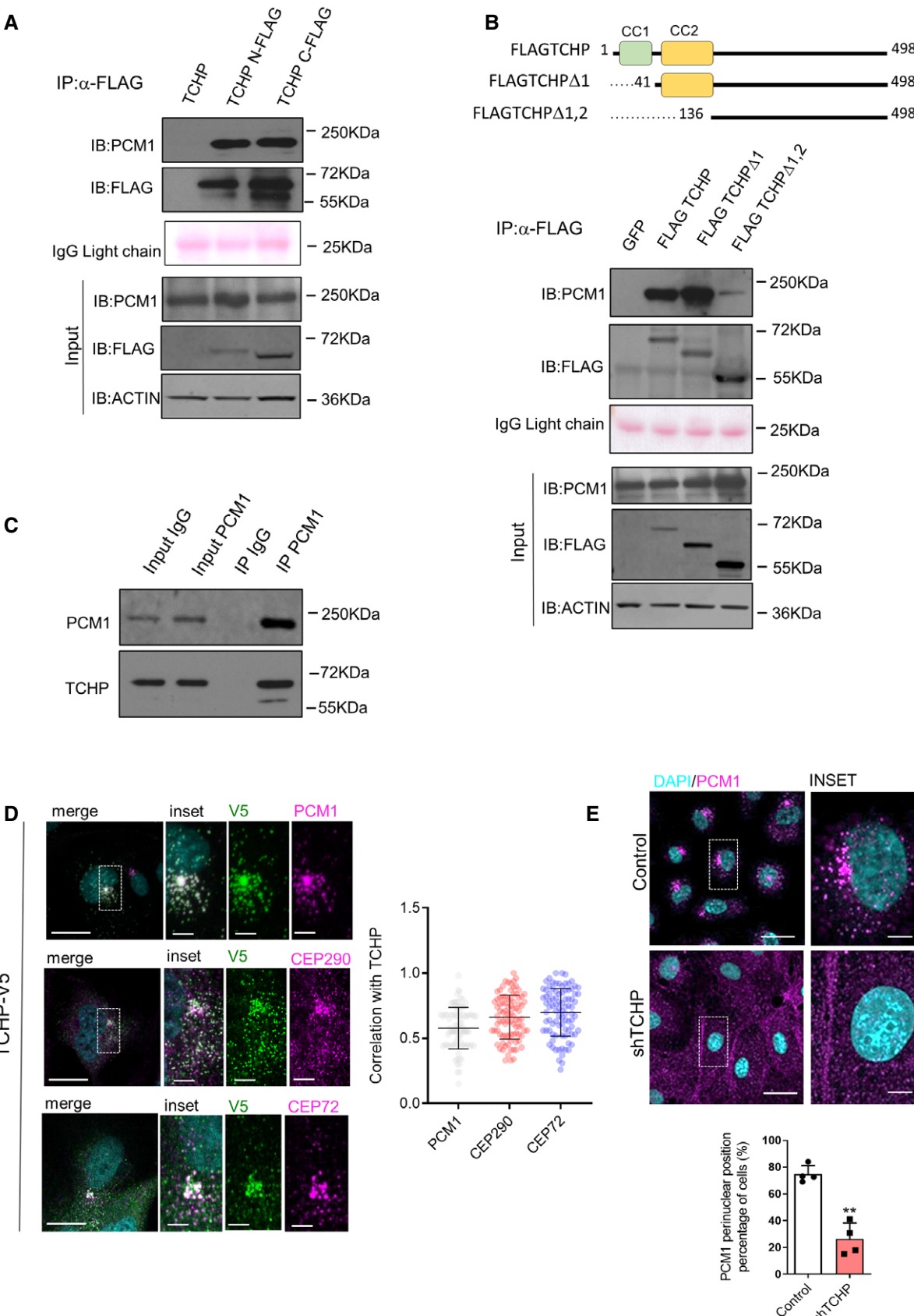

**Figure 1.**

## TCHP regulates PCM1 and GABARAP stability

Since the constant turnover of PCM1 is regulated by the proteolytic degradation [27], we analysed PCM1 protein levels and degradation rate in the TCHP-depleted cells. Like centrosomes and cytoskeleton-associated proteins [28], PCM1 was enriched in the Triton X-100 insoluble fraction, and TCHP depletion reduced PCM1 at the steady state (Fig 2A). Next, we used cycloheximide (CHX) and MG132 to block the translation and proteasomal degradation, respectively. Expression of PCM1 and GABARAP increased in TCHP knock-down cells compared to the control cells (Appendix Fig S2A and B). In TCHP-depleted cells treated with CHX, PCM1 protein degradation was enhanced compared to control cells, while inhibited by MG132, thus revealing the augmented proteasome-dependent turnover of PCM1 (Fig 2A).

PCM1 binds directly to GABARAP through a canonical LIR motif regulating GABARAP-specific autophagosome formation [10]. Although PCM1 depletion did not affect autophagy *per se*, it has destabilized GABARAP, but not LC3 [29], through proteasomal degradation [10]. Recently, the PCM1-GABARAP interaction has been further dissected through the analysis of the crystal structure of the PCM1 LIR motif bound to GABARAP, demonstrating that the manipulation of the key sites in either the PCM1 LIR motif or sequences flanking the LIR motif can alter the binding specificity of autophagy adaptors and receptors to ATG8 proteins [30]. The preference in PCM1 binding might explain the non-redundant functions of LC3 and GABARAP subfamilies.

GABARAP protein is essential for maturation and expansion [8] and autophagosome fusion with lysosomes [9]. Accordingly, we confirmed in ECs, a proteasome-dependent reduction of GABARAP during the 6 h of treatment with CHX. Without TCHP, in line with reduced stability of PCM1, the rate of GABARAP degradation was enhanced (Fig 2B). Conversely, ectopic expression of TCHP-V5 increased PCM1 (Fig 2C) and GABARAP protein levels (Fig 2D) extending their stability. Altogether, these data suggest that TCHP regulates PCM1 and GABARAP by proteasomal degradation. Whether TCHP affects the stability of other centriolar satellites components or whether the loss of TCHP had a more severe effect on the integrity of centriolar satellites remains to be determined.

## TCHP down-regulation impairs autophagic homeostasis

We next set to establish what role TCHP plays in autophagy. Transmission electron microscope (TEM) revealed a significant increase in the number of autophagic vesicles when TCHP is depleted in HUVECs (Fig EV2A).

Alongside the reduction in GABARAP, immunocytochemistry staining revealed an increased number of LC3- and p62-positive puncta (Fig 3A). We next set to determine the regulation of the autophagic flux by analysing the levels of p62 and LC3 in basal and Hank's buffered (HBSS) starved cells with or without bafilomycin A1 (BafA1). In a full medium, Western blot analysis confirmed an increased level of the lipidated form of LC3 (LC3-II) band and an increase of p62 protein levels in TCHP knock-down cells compared with control (Fig 3B). When autophagic flux was blocked with BafA1 at basal conditions, there was a higher accumulation of p62 in the control cells compared with TCHP knock-down cells. On the other hand, the LC3 lipidation rate increased more in control cell

than in the TCHP knock-down cells after autophagy stimulation. Finally, the treatment with HBSS re-activated the autophagic flux in TCHP knock-down cells as demonstrated by substantial degradation of LC3-II and reduction of p62 (Fig 3B).

The reduced autophagic flux was further analysed using mCherry-EGFP-LC3 assays as a complementary approach [31]. The mCherry fluorescence was lower in TCHP knock-down cells compared with the control, attesting to a decrease in autolysosome formation and a slower autophagic flux in cells lacking TCHP (Fig 3C). There was not a significant difference in the percentage or a total number of mature autolysosomes following starvation in HBSS medium (Fig 3C). These results were also confirmed by quantitative ratiometric flow cytometry analysis showing a decrease in mCherry/GFP fluorescence ratio in TCHP knock-down cells compared with the control, while HBSS treatment increased the ratio in both conditions (Fig 3D).

To further confirm the impairment of autophagic flux in TCHP knock-down cells, we performed a non-radioactive pulse-chase protocol using L-azidohomoalanine (AHA) labelling to quantify long-lived protein degradation during autophagy [32] (Figs 3E and EV2B). TCHP knock-down cells conserved a substantial amount of cellular fluorescence intensity as compared with the control sample. Conversely, after autophagy stimulation, under amino acid starvation for 2 h, both TCHP knock-down and control cells had reduced fluorescence intensity (Figs 3E and EV2B).

Finally, overexpression of TCHP reduced the basal level of p62 in ECs and the accumulation of p62 after BafA1 treatment under full growth medium (Fig 3F) and increased the degradation of p62 during nutrient starvation or treatment with torin-1 (Fig 3F). Moreover, exogenous TCHP partially rescued PCM1 localization in the centriolar satellites (Fig EV2C) and decreased p62 accumulation in ECs lacking TCHP (Fig EV2D). In an attempt to validate the rescue experiments using TCHP mutants, we observed that exogenously expressed TCHPΔ1,2, unlike wild-type TCHP or TCHPΔ1, failed to co-localize with PCM1, at least in HeLa cells. Moreover, a scattered PCM1 staining pattern was observed, similar to that seen after TCHP knock-down. The latter finding would suggest that the TCHPΔ,1,2 mutant could act as a dominant-negative for TCHP or perhaps other centriolar satellite proteins. Further studies will be required to elucidate the independent role of TCHPΔ,1,2 mutant in ECs. Overall, these results demonstrated a reduced autophagic flux in TCHP knock-down cells. Nevertheless, although the autophagic flux is reduced in TCHP knock-down cells, stress-induced autophagy appears to be functional, suggesting that TCHP-dependent reduction of basal autophagy is reversible and could be pharmacologically re-activated.

## The depletion of TCHP inhibits autophagosome maturation and efficient delivery of p62 to the lysosomes

Since GABARAP is critical for autophagosome expansion and maturation [9], we performed the proteinase K protection assay [33] to assess the efficiency of cargo receptor loading during autophagosome biogenesis and maturation. Autophagic vesicles were isolated by cytoplasm differential centrifugation and treated with proteinase K to determinate the proportion of the cargo receptor p62, and the ATG8 family proteins, LC3 and GABARAP, are not accessible to the protease because protected within autophagosome. We found that the

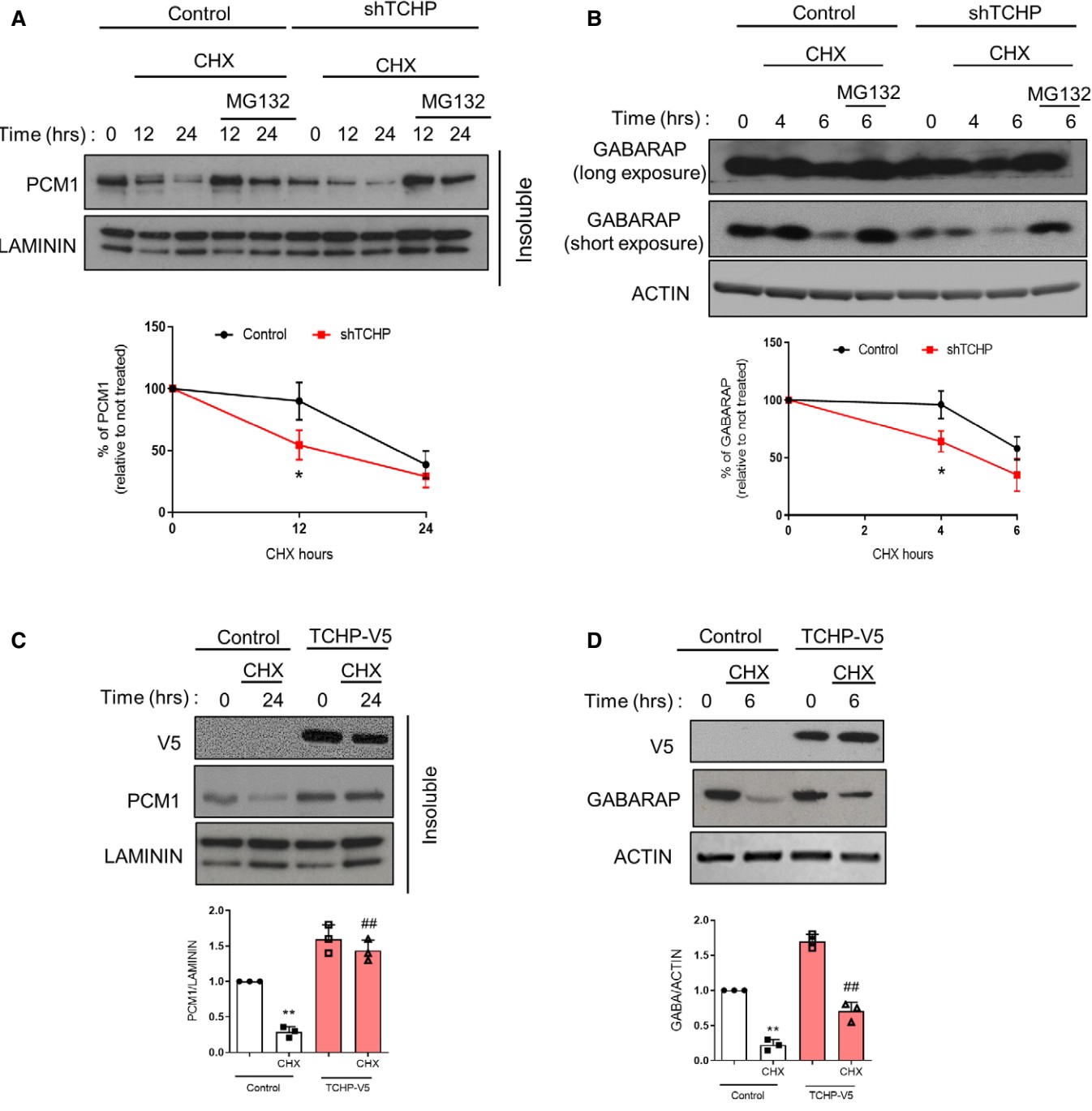

**Figure 2. TCHP regulates PCM1 and GABARAP stability.**

A, B (A) Control or TCHP knock-down HUVECs were subjected to cycloheximide (CHX) and MG132 treatments for the indicated number of hours before immunoblotting. PCM1 levels were analysed in the insoluble fraction. Laminin has been used as a loading control. *Below panel*: quantification of PCM1 degradation (one-way ANOVA; *P = 0.0180 vs. control time 0). (B) Condition as in (A), Western blot for anti-GABARAP and anti-ACTIN antibodies. *Below panel*: quantification of GABARAP degradation (one-way ANOVA; *P = 0.0215 vs. control time 0).

C, D (C) Control or TCHP overexpressing HUVECs were subjected to CHX treatment for the indicated number of hours prior to immunoblotting. Western blot was probed for anti-PCM1 and anti-V5 antibodies. Laminin has been used as a loading control. (D) Condition as in (C), Western blot was probed for anti-V5, anti-GABARAP and anti-ACTIN antibodies. *Below panels*: quantification of (C) (one-way ANOVA; **P < 0.0001 vs. control time 0; ##P = 0.0003 vs. control CHX) and (D) (one-way ANOVA; **P < 0.0001 vs. control time 0; ##P = 0.0056 vs. control CHX).

Data information: Statistical analyses were performed on at least three independent experiments. Data are represented as mean ± SD.
Source data are available online for this figure.

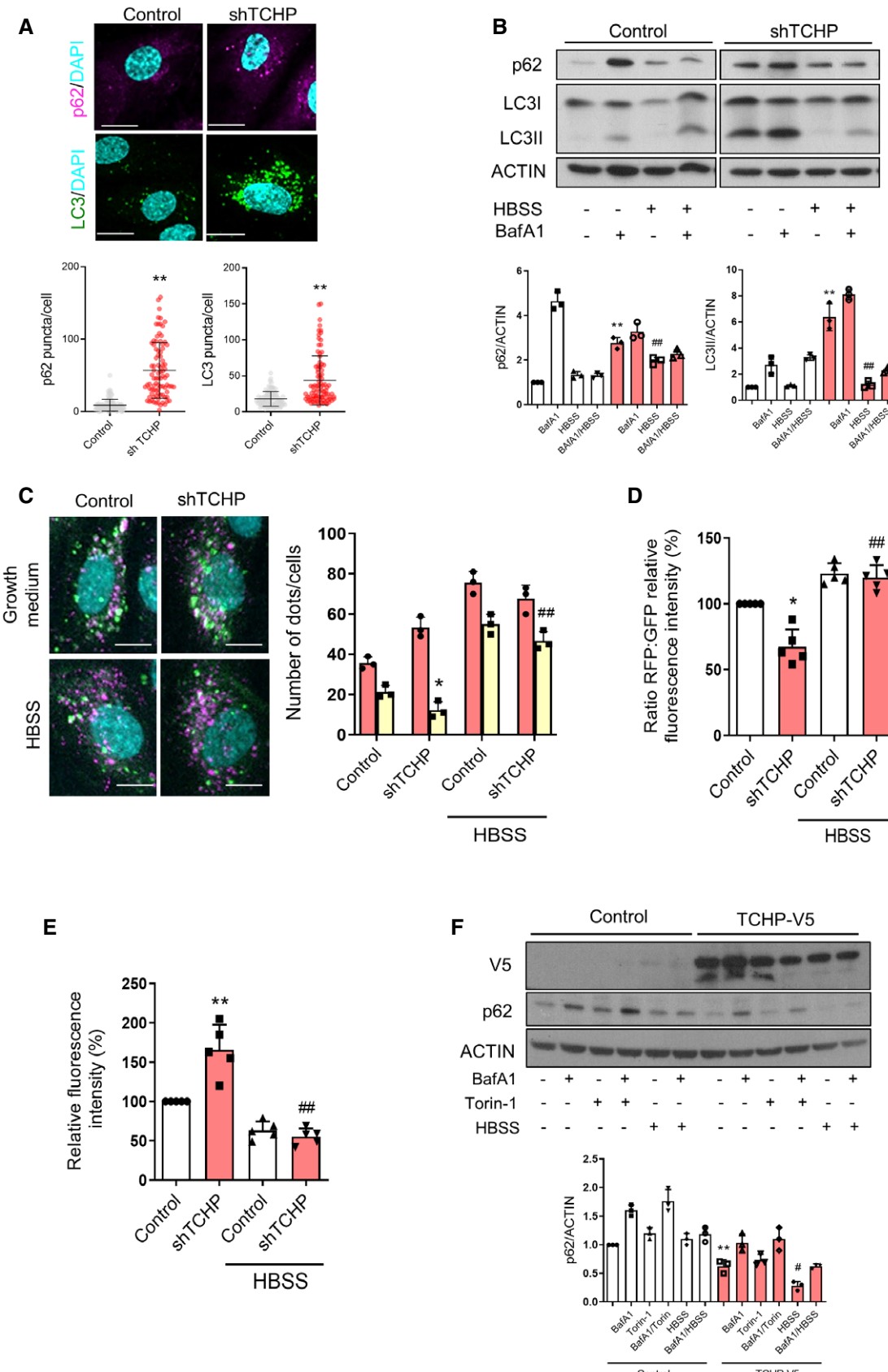

**Figure 3.**

**Figure 3. Analysis of autophagy in TCHP-depleted endothelial cells.**

A Immunofluorescent staining for LC3 and p62 in TCHP knock-down and control cells. Scale bars, 25 μm. Lower panel: quantification (*n* = 90 cells, unpaired *t*-test; LC3: **P < 0.0001 vs. control; p62: **P < 0.0001 vs. control).

B Western blot of p62 and LC3 during under normal culture condition or starved condition (HBSS) or the presence of BafA1 (2 h) in TCHP knock-down or control cells. *Lower panels:* p62 quantification (one-way ANOVA; **P = 0.0003 vs. control, ##P = 0.0093 vs. shTCHP) and LC3 quantification (one-way ANOVA; **P = 0.0008 vs. control, ##P = 0.0011 vs. shTCHP).

C HUVECs were transduced with the tandem mCherry-EGFP-LC3 and with shRNA TCHP or control vectors. *Left panels*: representative picture of mCherry-EGFP-LC3 reporters. Scale bars, 25 μm. *Right panel*: quantification of the number of mCherry-only (red bars, autolysosomes) or double-positive (mCherry+/EGFP+; yellow bars, autophagosomes) (*n* = 80 cells, one-way ANOVA; *P = 0.0447 vs. control; ##P = 0.0007 vs. shTCHP).

D Ratiometric flow cytometric analysis of mCherry-EGFP-LC3 reporters as in C (one-way ANOVA; *P = 0.0247 vs. control; ##P = 0.0088 vs. shTCHP).

E Quantification of long-lived protein degradation assay in TCHP knock-down and control cells by flow cytometry (one-way ANOVA; **P = 0.0018 vs. control, ##P < 0.0001 vs. shTCHP).

F HUVECs were transduced with TCHP-V5 and control vectors. Western blot for anti-V5 and anti-p62 antibody during under normal culture condition or HBSS or BafA1 or torin-1 treatment. *Lower panel*: quantification (one-way ANOVA; **P = 0.0044; #P = 0.0117 vs. TCHP-V5).

Data information: Statistical analyses were performed on at least three independent experiments. Data are represented as mean ± SD.

sensitivity of p62, LC3 and GABARAP to proteinase K was enhanced by TCHP knock-down in the low-speed pellet (LSP) and high-speed pellet (HSP) in full medium and after BafA1 treatment (Fig 4A).

We then analysed the structure of autophagosomes in control and TCHP knock-down cells treated with BafA1 (Fig 4B). Accordingly, with documented roles for ATG8s and in particular GABARAP family in enlarging autophagosomal membranes [34], we observed a considerable reduction for the average size of autophagic vesicles (AVs) and autophagosomes (APs) in TCHP knock-down cells (Fig 4B).

Since syntaxin 17 (STX17) is localized explicitly at mature autophagosomes [35], we analysed the recruitment of STX17 to autophagosome to distinguish unsealed and immature phagophore from mature autophagosomes. We observed a significant decrease in the number of cells with double-positive for LC3 and STX17 in TCHP knock-down cells at the basal condition and after BafA1 treatment (Fig 4C). Moreover, STX17 recruitment to the autophagosome (LC3/STX17-positive puncta per cells) is disrupted in TCHP knock-down cells during the BafA1 treatment (Fig 4C).

Collectively, these results demonstrated that TCHP down-regulation impairs basal autophagy leading to the accumulation of unresolved autophagosomes. These conclusions, however, do not exclude possible direct participation of TCHP in autophagosome maturation. Recent studies in the transition from phagophore to autophagosome are expanding the number of proteins and molecular machineries, such as TRAPP [36] and CHAMP2A [37], operating at the autophagosome closure, highlighting its complexity.

We then examined the subcellular distribution of p62 in ECs stained for LAMP2, aggregates and ubiquitinated proteins. Knock-down of TCHP increased the percentage of p62 puncta that were also negative for LC3 and LAMP2 (Fig EV3A), therefore, and suggests that TCHP could impair the delivery of p62 to the lysosome (LAMP2-negative p62 puncta). Finally, in TCHP knock-down cells, a considerable fraction of p62 puncta failed to co-localize with the aggregates and ubiquitinated proteins (Fig EV3B and C).

## Lack of TCHP affects lysosome distribution but not the lysosomal activity

Although we observed that lack of TCHP affected autophagic flux and autophagosome maturation, we then analysed whether the lysosomes activity or distribution are compromised in TCHP knock-down cells.

Immunostaining demonstrated a discrete alteration in shape, distribution and intensity of RAB11, EEA1 and RAB7 vesicles (Fig EV4A). Besides, knock-down of TCHP altered the cellular positioning of lysosomes as shown by LAMP2 immunolabelling, inducing a marked perinuclear clustering of these organelles. Further, increased intensity of LysoTracker Red™ in TCHP knock-down cells suggested increased acidification and activity of lysosomes (Fig EV4B).

We then performed the epidermal growth factor (EGF) receptor (EGFR) degradation assay to analyse lysosome activity [38]. Endocytosis and subsequent lysosomal-mediated degradation are the primary regulators of EGFR stability following ligand activation. Cell stimulation with EGF upon knock-down of TCHP resulted in an increasing degradation of EGFR compared with control (Fig EV4C).

We then analysed the lysosome distribution and microtubule (MT) network in TCHP knock-down cells and control cells during

**Figure 4. TCHP contributes to autophagosome maturation.**

A The postnuclear fraction (PNS) from TCHP knock-down and control HUVECs in the presence or absence of BafA1 was separated into (low-speed pellet) LSP and (high-speed pellet) HSP fractions and then analysed by immunoblots using anti p62, LC3 and GABARAP antibodies. The sub-fractions were treated with proteinase K (Prot. K) with or without Triton X-100 (TX-100). Quantification of the Western blot: the p62, LC3 and GABARAP levels relative to respective GAPDH were quantified using densitometry analysis and normalized to the value of non-treated samples.

B Representative TEM images of autophagosomes in control and TCHP knock-down HUVECs in the complete medium after 3-h incubation with BafA1 (scale bars, 600 nm): (i), quantification of mean autophagic vesicles (AVs) and autophagosome (APs) area and (ii), distribution of the cross-section areas of the analysed vesicles expressed in percentage (*n* = 7 cells; > 100 vesicles per sample; AVs: unpaired *t*-test; **P < 0.0001 vs. control; APs: unpaired *t*-test **P < 0.0001 vs. control).

C *Left panels:* representative single-channel and merged images of HUVECs expressing GFP-LC3 and immunostained for STX17. Scale bars, 25 and 2 μm in the inset. *Right panels*: quantification of LC3/STX17-positive cells in the presence or absence of BafA1 (one-way ANOVA; **P = 0.0060 vs. control; #P = 0.0136 vs. control BafA1) and quantification LC3/STX17-positive puncta per cells (*n* = 50 cells; one-way ANOVA; **P < 0.0001 vs. shTCHP BafA1).

Data information: Statistical analyses were performed on at least three independent experiments. Data are represented as mean ± SD.

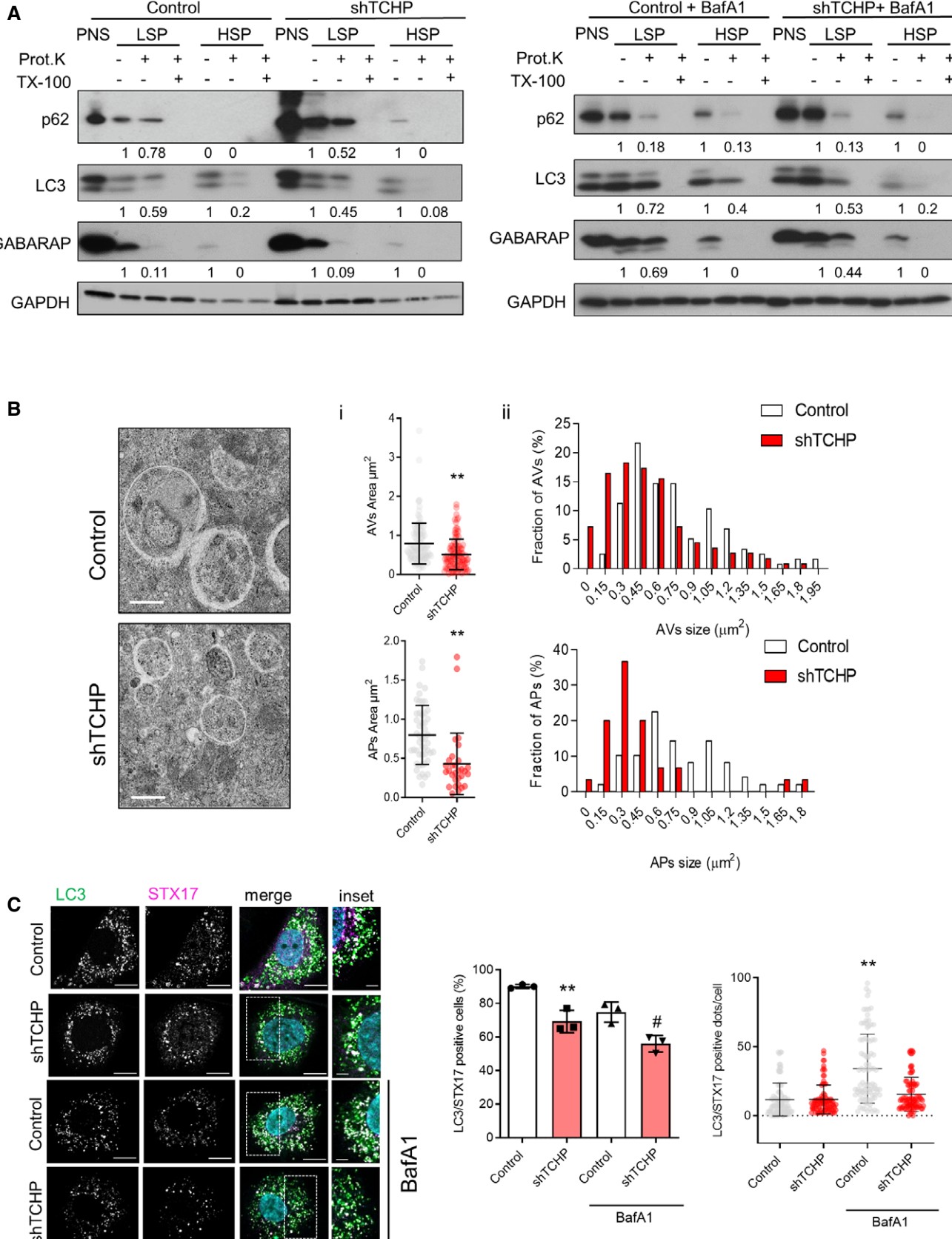

**Figure 4.**

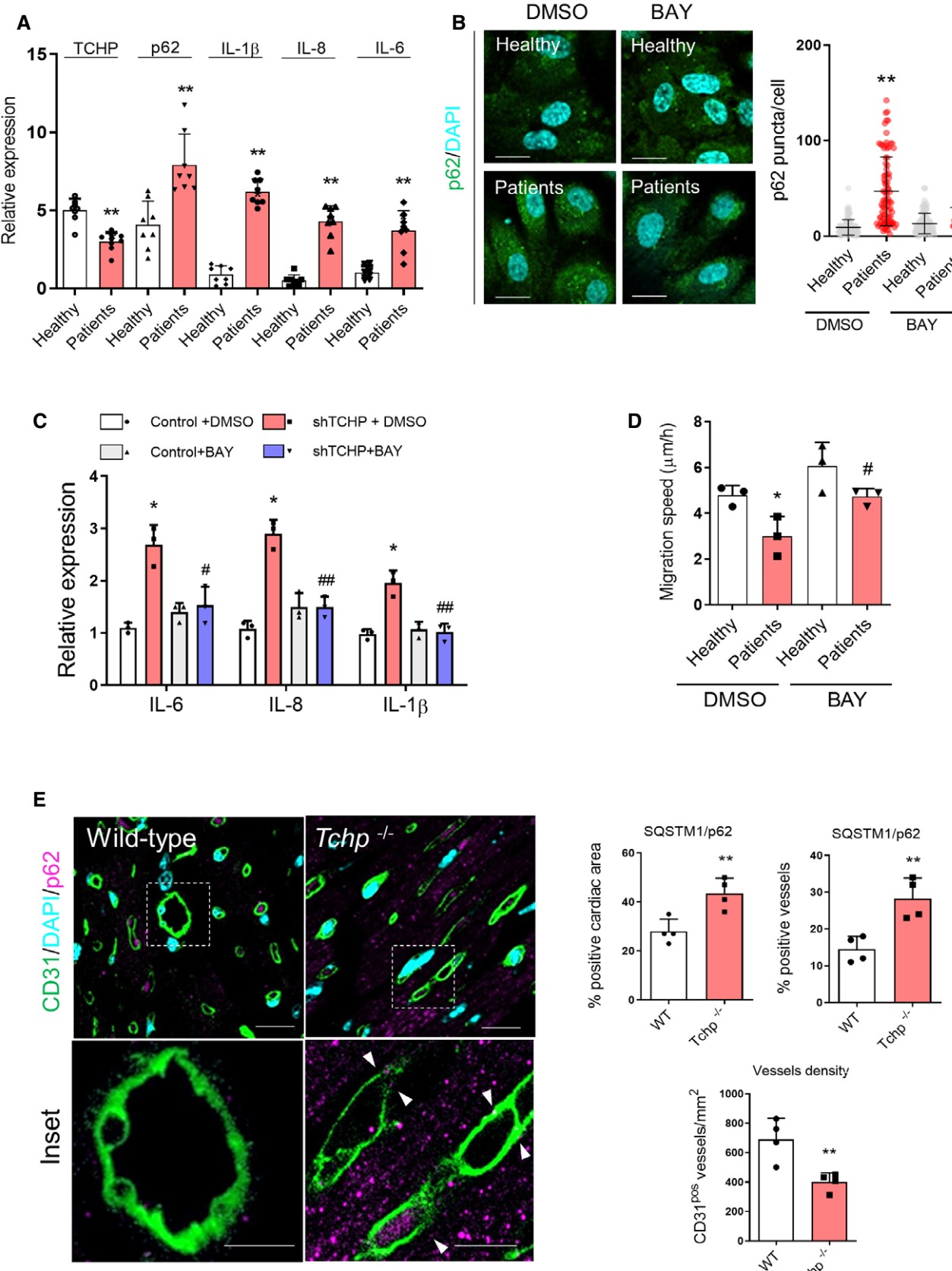

**Figure 5.**

◀

**Figure 5. p62 accumulation in ECs from patients with CAD and Tchp knock-out mice.**

A Expression of TCHP, p62, IL-1β, IL-8 and IL-6 in ECs from vessels wall from patients with CAD ($n$ = 8 patients per group, unpaired $t$-test; TCHP **$P < 0.0001$; p62 **$P = 0.0007$; IL-1β **$P < 0.0001$; IL-8 **$P < 0.0001$; and IL-6 **$P < 0.0001$ vs. healthy subject).

B Staining and quantification of p62 in ECs from healthy subject and patients either treated with vehicle (DMSO) or BAY. Scale bars, 50 μm; ($n$ = 3 patients per group; $n$ = 80 cells; one-way ANOVA; **$P < 0.0001$ vs. healthy DMSO; ##$P = 0.003$ vs. patients DMSO).

C Expression of IL-6, IL-8 and IL-1β ($n$ = 3 patients per group; one-way ANOVA; IL-6: *$P = 0.0135$ vs. healthy DMSO; #$P = 0.0180$ vs. patients DMSO; IL-8: *$P = 0.0139$ vs. healthy DMSO; ##$P = 0.0024$ vs. patients DMSO; IL-1β: *$P = 0.0104$ vs. healthy DMSO; ##$P = 0.0069$ vs. patients DMSO).

D Migration speed, in ECs from healthy subject and patients either treated with vehicle (DMSO) or BAY ($n$ = 3 patients per group; one-way ANOVA; *$P = 0.0325$ vs. healthy DMSO; #$P = 0.0336$ vs. patients DMSO).

E *Left panels*: representative images of the heart of wild-type and Tchp knock-out mice stained for CD31 (green) and p62 (magenta). Scale bars, 100 and 50 μm for the inset. *Right panels*: quantification of the percentage of cardiac area ($n$ = 4 mice per group; unpaired $t$-test; **$P = 0.083$ vs. wild-type) or vessels positive for p62 ($n$ = 4 mice per group; unpaired $t$-test; **$P = 0.058$ vs. wild-type) and vessel density ($n$ = 4 mice per group; unpaired $t$-test; **$P = 0.0097$ vs. wild-type).

Data information: Data are represented as mean ± SD.

starvation (endothelial basal medium without FBS) and recovery (replenishment of serum). Analysis of lysosome distribution showed that, in control cells, the starvation increased the proportion of cells with predominantly perinuclear lysosomes, while upon recovery, LAMP2-positive vesicles localized at the cell periphery (Fig EV4D). On the contrary, TCHP knock-down cells showed that the perinuclear distribution of lysosome in growing condition (full medium) is not changed during starvation or recovery conditions (Fig EV4D). Previous studies reported that TCHP is required for functional MT anchoring to the centrosome [16]. As expected, TCHP knock-down cells showed an unfocused MT network in growing conditions which it is not modulated by starvation (Fig EV4D). Altogether, our results have revealed that the loss of TCHP inhibits the lysosome distribution progressively without substantially changing the lysosomal acidity and activity. Finally, p62 level in TCHP knock-down cells decreased in serum starvation conditions as in control cells (Fig EV4E), confirming that in TCHP knock-down cells, the autophagic process is not affected by the limited mobility of lysosomes and the defective MT network.

### NF-κB inhibition reduces p62 accumulation, restoring endothelial cell function

To elucidate the mechanisms behind the accumulation of p62 in ECs, we have developed a phenotypic screening assay to select compounds, which could decrease the accumulation of p62 in ECs lacking TCHP. We screened a library of 176 target-annotated compounds (including protease, epigenetics and kinase inhibitors) at two different doses (0.3 and 3.3 μM) with the primary outcome to reduce the number of p62 cytoplasmic puncta (Appendix Fig S3A and B). The screen generated a list of 25 hits (Table EV2) with the top ones being excluded due to high toxicity (apicidin and terreic acid) or negative effect on endothelial function (splitomicin) [39]. The screen identified BAY11-7082 (IKK inhibitor), TYRPHOSTIN AG1288 (tyrosine kinases inhibitor) and SB202190 (p38MAPK inhibitor) as having a definite effect on reducing p62 accumulation in TCHP knock-down cells (Fig EV5A); named compounds were further tested in secondary functional assays. BAY11-7082 was the only compound that was able to reduce cytokine transcription (Fig EV5B) and restore migratory capacity in TCHP knock-down ECs (Fig EV5C). TCHP knock-down cells exhibited a marked increase in NF-κB S536 phosphorylation compared to the control cells. Treatment with BAY11-7082 reduced NF-κB phosphorylation (Fig EV5D) and p62 expression

(Fig EV5E) in TCHP knock-down cells, revealing an NF-κB-dependent contribution to p62 accumulation. As expected, silencing of NF-κB/p65 reduced p62 expression in ECs lacking TCHP (Fig EV5F). Since NF-κB response element has been identified in the p62 promoter [40,41], TCHP knock-down ECs showed an increased enrichment of NF-κB/p65 on IκBα (as positive control locus for NF-κB translocation [42]) and p62 promoters by chromatin immunoprecipitation (Fig EV5G).

P62 was initially described as a scaffold protein and a signalling hub for the interactions with many types of enzymes through different binding domains [43]. It is well known also that p62 induces inflammatory cytokines production via TRAF6 polyubiquitination and thereby NF-κB activation [44]. Instead, a previous study demonstrated that stimulated autophagy, by enhanced degradation of p62, reduced inflammation, whereas blocking autophagy had an opposite effect [45]. Thus far, the phenotypic screens identified genes which reduce the accumulation of p62 following stress stimuli, as a novel approach to map autophagy pathways [46,47]. While our screen was designed to identify compounds targeting the accumulation of p62, it is likely that among our hits, we also identified compounds which regulate autophagy.

### Accumulation of p62 in ECs of patients with premature coronary artery disease and the heart and vessels of Tchp knock-out mice

Accumulation of p62-positive aggregates is among the best-known characteristics of autophagy-deficient tissues [11]. We analysed the expression of TCHP, p62 and cytokines in the ECs from patients with premature coronary artery disease (CAD). The ECs were obtained from the vessel wall of patients with endothelial dysfunction, comprising significant impairments in proliferation, adhesion and migration [48]. Gene expression analysis showed that ECs from patients express a low level of TCHP and high level of p62 and cytokines in comparison with the ECs from healthy donors (Fig 5A). Treatment of ECs from patients with BAY11-7082 has reduced the accumulation of p62 puncta (Fig 5B) and cytokine expression (Fig 5C), while it improved their migratory capacity (Fig 5D). Being that NF-κB is a critical mediator of endothelial cell dysfunction and impairs vascular regeneration [49], reactivation of autophagy through inhibition of NF-κB may help to restore vascular function and reparative angiogenesis.

Heart of Tchp knock-out mice presented a reduced cardiac vascularization with a significant accumulation of p62 in the cardiomyocytes and the vessels (Fig 5E). Finally, p62 accumulated in the

liver and pancreas of Tchp knock-out mice (Appendix Fig S4). Together, our data ascertain the secure link between depletion of TCHP, p62 accumulation and vascular function.

In conclusion, these results reveal for the first time the pivotal role for TCHP in linking EC function with the control of basal autophagy, highlighting a possible novel role in vascular disease.

# Materials and Methods

### Cells and cell culture and reagents

Human umbilical vein ECs (HUVECs) and ECs from healthy donor and patients were cultured in EGM-2 (EBM-2 + SingleQuots™ Kit) and 2% foetal bovine serum (FBS) (Lonza). HUVECs and ECs were used between P2 and P6 passages. HEK293T (ATCC) were grown in Dulbecco's modified Eagle's medium (DMEM) with 4.5 g/l glucose, 2 mM L-glutamine, without Na pyruvate (Lonza), 10% FBS and 1% penicillin/streptomycin (Pen/Strep). *Reagents and doses were as follows*: LysoTracker (Thermo Fisher), bafilomycin A1 (Sigma, 200 nM), torin-1 (CST, 10 μM), BAY11-7082 (Abcam, 300 nM), TYRPHOSTIN AG1288 (Abcam, 300 nM), SB202190 (Tocris, 300 nM), cycloheximide (Sigma, 2 μM) and MG132 (Sigma, 2 μM).

### Lentiviral vectors, plasmid constructs and siRNAs

The pLKO DNA plasmids containing the shRNA sequence against human TCHP were purchased from Sigma-Aldrich (Mission®RNAi TRCN0000127662 and TRCN TRCN0000130868). The scrambled sequence shRNA plasmid was purchased from Addgene, plasmid #1864. The packing plasmids used were pCMV-dR8.2 dvpr, plasmid #8455 and pCMV-VSVG, plasmid #8454 from Addgene. p62-V5 (HsCD00434166), TCHP-V5 (HsCD00444989) and empty vector (pLX304) are from DNASU plasmid collection, and pBABE-puro mCherry-EGFP-LC3B (Addgene plasmid # 22418) and pBABEpuro GFP-LC3 (Addgene plasmid # 22405) were a gift from Jayanta Debnath. To generate an expression plasmid for 3xFLAG-tagged, the full-length TCHP coding sequence was amplified by PCR with primers having the sequence 5′-GATGACAAGCTTGGAAACTCCG AGCCTCAGAGA-3′ and 5′ GGATCCTCTAGATTCTCTGTACTTATG GTACCC-3′. The PCR product was digested with HindIII and XbaI, and the resulting DNA fragment was inserted into p3xFLAG-CMV-7.1 (Sigma) to prepare p3xFLAG-TCHP. A 3xFLAG–TCHP coding sequence was then amplified by PCR; the products were gel purified and verified by sequencing. The deletion mutants FLAG-TCHP Δ1, lacking the first coiled motif of TCHP protein, and FLAG-TCHP Δ1,2, lacking the first and second coiled-coil motives, were generated by PCR starting from the previously described p3XFLAG-TCHP full-length construct. The products of the PCR were gel-purified, verified by sequencing and cloned into the HindIII-XbaI sites of the p3x-FLAG-CMV-7.1 expression vector.

The primers used in PCR are TCHP Δ1 Fw_: 5′-GCGAT TAAGCTTTTCAGGATGTCTGACATCTGC-3′; TCHP Δ1.2 Fw_: 5′-G CGATTAAGCTTCAACTTTTGTACGAACACTGG-3′; and TCHP Rw_: 5′-GGATCCTCTAGATTCTCTGTACTTATGGTACCC-3′.

siRNA for NFκBp65 is SMARTpool: ON-TARGETplus RELA siRNA (Dharmacon), and siRNA for TCHP is SMARTpool: ON-TARGETplus TCHP siRNA (Dharmacon).

### Endothelial cells functional assays

The following functional assays were performed as previously described [49]: Matrigel assay with HUVECs was performed using BD Matrigel Basement Membrane Matrix (BD Biosciences). Migration was analysed with the ECIS machine [ECIS chip array (8W1E)] (Applied Biophysics). The migration speed was calculated in micrometres per hour.

### Proteinase K protection assay

The subcellular fractionation (PNS; postnuclear supernatant = $300 \times g$ for 5 min at 4°C, LSP; low-speed pellet = $700 \times g$ for 5 min at 4°C, HSP; high-speed pellet, HSS; high-speed supernatant = $100,000 \times g$ for 30 min at 4°C) of control of TCHP knock-down ECs was performed as described in Ref [33]. In brief, each fraction of LSP and HSP was treated with 100 μg/ml proteinase K on ice with or without 0.5% Triton X-100 for 30 min. The fraction samples were precipitated with 10% trichloroacetic acid, washed with ice-cold acetone three times, resuspended in sample buffer including 3 M urea and then analysed by Western blot for p62, LC3, GABARAP and GAPDH antibody. GAPDH was used as a loading control, as described in Ref [33].

### *In vivo* Matrigel plug assay

Experiments involving mice were covered by the project and personal licenses issued by the UK Home Office, and they were performed in accordance with the Guide for the Care and Use of Laboratory Animals (the Institute of Laboratory Animal Resources, 1996) and in accordance with Animal Research Report of *In vivo* Experiments (ARRIVE) guidelines. CD-1 mice (male, 10 weeks old) were subcutaneously injected into the groin regions with 400 μl Matrigel containing recombinant mouse basic FGF (PeproTech, 250 ng/ml) and heparin (Sigma, 50 U/ml) mixed with control or TCHP siRNA (Dharmacon) [lipids (Lipofectamine RNAiMAX reagent, ratio 1:1 in volume) 5 μg/gel, $n = 5$ per group]. After 21 days, mice were sacrificed, and the Matrigel plugs were removed and fixed in 4% paraformaldehyde.

### RNA extraction and quantitative real-time analysis

Total RNA was extracted using miReasy kit (Qiagen). For mRNA analysis, cDNA was amplified by quantitative real-time PCR (qPCR) and normalized to 18S ribosomal RNA. Each reaction was performed in triplicate. Quantification was performed by the $2^{-\Delta\Delta Ct}$ method [50]. Primers are from Sigma (KiCqStart™ Primers).

### Chromatin immunoprecipitation

Nuclei were isolated from formaldehyde (1% final)-fixed HUVECs by lysing in ChIP Lysis Buffer [1% SDS, 10 mM EDTA, 50 mM Tris–HCL (pH 8.1)] supplemented with protease inhibitors. Chromatin was fragmented by sonication using a Bioruptor UCD-300 ultrasound sonicator (Diagenode). DNA-cross-linked proteins were immunoprecipitated (1% kept as input) using 5 μg of NF-κB/p65 (Millipore) or control mouse IgG antibody. The antibody was pulled down with protein G beads (Dynabeads, Thermo Fisher) at 4°C

overnight. Associated DNA was then purified by extraction using Monarch PCR & DNA Cleanup Kit (New England Biolabs). Immuno-precipitated DNA and total input were used as a template for real-time qPCR. ChIP primers for NF-κB/p65 on IκBα promoter are Fw GTGCGCCCTCAACTAACAGT and Rev CATCCCAATGAAGCTT CTGA. Identification of the NF-κB/p65 binding sites in the p62 promoter was performed retrieving ENCODE dataset GSM935527 (Chr 5: 179,245,409-179,256,832). Primers for NF-κB/p65 on p62 promoter are Fw CTAAAGATGGCCCAGAGCAG and Rev CCC CCTCCCAAATAATCCTA.

## Mass spectrometric analysis

Gel bands were subjected to overnight trypsin digestion, and peptide extracts were dried by Speedvac. The dried peptide samples were resuspended in MS-loading buffer (0.05% trifluoroacetic acid in water) and then filtered using Millex filter before HPLC-MS analysis. Nano-ESI-HPLC-MS/MS analysis was performed using an online system consisting of a nano-pump (Dionex Ultimate 3000, Thermo Fisher) coupled to a QExactive instrument (Thermo Fisher) with a precolumn of 300 μm × 5 mm (Acclaim Pepmap, 5 μm particle size) connected to a column of 75 μm × 50 cm (Acclaim Pepmap, 3 μm particle size). Samples were analysed on a 90-min gradient in the data-dependent analysis (1 survey scan at 70k resolution followed by the top 10 MS/MS). The gradient between solvent A (2% acetonitrile in water 0.1% formic acid) and solvent B (80% acetonitrile-20% water and 0.1% formic acid) was as follows: 7 min with buffer A, over 1 min increase to 4% buffer B, 57 min increase to 25% buffer B, over 4 min increase to 35%, over 1 min increase to 98% buffer B and stay under those conditions for 9 min, switch to 2% buffer B over 1 min and the column was conditioned for 10 min under those final conditions. MSMS fragmentation was performed under nitrogen gas using high energy collision dissociation in the HCD cell. Data were acquired using Xcalibur ver 3.1.66.10. Data from MS/MS spectra were searched using MASCOT Versions 2.4 (Matrix Science Ltd, UK) against the human subset of UniProt database with the maximum missed-cut value set to 2. Following features were used in all searches: (i) variable methionine oxidation, (ii) fixed cysteine carbamidomethylation, (iii) precursor mass tolerance of 10 ppm, (iv) MS/MS tolerance of 0.05 amu, (v) significance threshold ($P$) below 0.05 (MudPIT scoring) and (vi) final peptide score of 20. Progenesis (version 4 Nonlinear Dynamics, UK) was used for LC-MS label-free quantitation. Only MS/MS peaks with a charge of 2+, 3+ or 4+ were taken into account for the total number of "Feature" (signal at one particular retention time and m/z), and only the five most intense spectra per "Feature" were included. Results were exported using a peptide score cut off of 20. From the exported results sheet, differentially expressed proteins were considered significant if the $P$-value was less than 0.05 and if the number of peptides used in quantitation per protein was equal to or more than 2.

Protein network analysis has been performed using STRING software (https://string-db.org/).

## Phenotypic screening assay

### Image acquisition

Plates were imaged on a wide-field Imagexpress Micro XL high content microscope (Molecular Devices). Images of Hoechst-labelled nuclei and p62 antibody labelling were imaged using the DAPI and Cy3 filter sets from 4 different sites within the well using a 20× S Plan Fluor objective containing up to 200 cells per field of view.

### Image analysis

Images were analysed using a custom workflow developed in the MetaXpress Custom Module Editor (Molecular Devices). A top hat filter (size = 10 pixels, shape = circle) was applied to remove background fluorescence from the p62 images. Using the granularity module, nuclei were first identified in the DAPI image using a user-defined intensity above local threshold method and maximum (30 μm) and minimum (10 μm) widths. p62 puncta were then detected by using a user-defined intensity above local background and maximum (1 μm) and minimum (0.5 μm) widths. The nuclear objects were then used as seeds to create a pseudo-whole cell by growing the mask by 50 pixels. Finally, the nuclear region was subtracted from the whole-cell mask to give nuclear and cytoplasmic masks. The number of p62 puncta in the nuclear and cytoplasmic regions was counted per cell.

### Data analysis

Data handling and analysis was done using Spotfire High Content Analyser software (PerkinElmer). Data were aggregated to whole well averages and were plate normalized to the negative controls by dividing all values on the plate by the median value for the negative control and then scaling the values between 0 and infinity, with 1 being the median of the negative controls on that plate. For hit identification, control cells (no shRNA) were used as positive controls and an ensemble-based tree classifier was used to identify hit compounds with a cross-validated fivefold CV misclassification error of 0.4%. The following features were used in the ensemble-based tree classifier: Cell_Count;Nuclear_Granules_Nuclear_count_Sum; Cytoplasmic_Granules_Nuclear_Count_Sum.

## Western blot analyses

Total proteins were extracted in RIPA buffer containing 1 mM sodium orthovanadate and Complete Protease Inhibitor Cocktail (Roche Applied Science) and quantified using the Pierce™ BCA protein assay kit (Thermo Fisher). Equal amounts of proteins were loaded onto SDS–polyacrylamide gels and transferred to PVDF membrane. The membranes were then blocked with 5% non-fat milk in TBST 0.1% and immunoblotted overnight at 4°C with the following primary antibodies (1:1,000): TCHP (Santa Cruz Biotechnology, SC-515025), β-actin (Sigma, A5441), p62 (GeneTex, GTX100685), LC3B (CST, #2775), p16 (BD Biosciences, 551154), NF-κB (Millipore, MAB3026), (S536) NF-κB (CST, #3033), PCM1 (CST, #5213) and GABARAP (CST, #13733), NDP52 (CST, #60732) and V5 (Thermo Fisher, R960-25). Secondary antibodies (1:5,000): anti-Mouse IgG–Peroxidase (Sigma, A5906) and anti-Rabbit IgG–Peroxidase (Sigma, A0545), were incubated for 1 h at RT. Pixel intensity/quantification was performed using ImageJ.

## Immunofluorescence

HUVECs cells were plated on fibronectin-coated glass coverslips. Twenty-four hours later, the slides were fixed with 4% paraformaldehyde, permeabilized with 0.05% Triton X-100 in PBS

and then incubated with the primary antibody (1:400) in 3% BSA overnight at 4°C. Secondary antibodies diluted 1:1,000 in 3% BSA. Slides were imaged on Zeiss LSM-780 confocal. Primary and secondary antibodies used for immunofluorescence were as follows: PCM1 (CST, #5213), p62 (GeneTex, GTX100685), LC3B (CST, #2775), Rab7 (CST, #9367), Rab11 (CST, #5589), EEA1 (CST, #3288), LAMP2 (CST, #49067), STX17 (Proteintech, 17815-1-AP), CEP290 (Proteintech, 22490-1-AP), CEP72 (Proteintech, 19928-1-AP) and α-Tubulin (Abcam, ab184613). Aggregates of ECs were measured using the PROTEOSTAT kit (Enzo Life Science, ENZ-51035-0025). Senescence of ECs was measured using the Cellular Senescence Assay Kit from Cell Biolabs, Inc. (Cat: CBA-230).

## Imaging analysis

The image analysis software Cell Profiler was used to quantify co-localization and lysosome distribution. A detailed description of each stage of the Cell Profiler workflow is located on their website (http://cellprofiler.org).

### Cell profiler workflow for co-localization
The images were processed and subjected to a 20-module co-localization workflow. To summarize, the images are initially loaded as separate channels, corrected for light illumination and aligned. The images then undergo pixel-based correlation, where the pixel intensities are compared in each image and any initial correlation determined. The specific structures of interest are determined by thresholding the images and segmenting them into objects, this allows for any co-localization to be determined between the individual channels and objects of interest. During the final stage of image analysis, the images are further enhanced, and the objects of interest are refined. The software then calculates various statistics within the defined regions and calculates the number of pixels within a specific object in each channel. The area occupied by co-localized regions is then divided by the area of co-localized objects of interest and a per image pixel fraction determined from the total object pixels. This information is then converted into a percentage co-localization or Pearson correlation and the data exported.

### Cell profiler workflow for lysosome distribution
Radial integrated fluorescence intensity was measured by using eight concentric circles centred on the nucleus area (each nucleus then acted as a seed for subsequent 2 μm rings to be draw radiating from its periphery), thus covering 16 μm radial distribution of individual cells. The sum of fluorescence integrated intensity of the inner four circles was considered as perinuclear and the sum of the outer four circles as the cytoplasm. Then, the percentage of perinuclear localization was calculated by dividing the sum of perinuclear fluorescence by total fluorescence ([perinuclear/[perinuclear + cytoplasm]] × 100). Each treatment was evaluated by quantifying 20 cells per condition and per experiment.

### Microtubule density
The intensities of microtubule network were quantified within manually defined areas tracing the microtubule ends. Images were imported into ImageJ software where the cell periphery of cells was identified using a thresholding method. The total pixel density/

mean fluorescence intensity per whole cell after subtraction of the background was measured and plotted for each condition.

## Electron microscopic analysis

Cell pellets were prepared and fixed in 4% paraformaldehyde or 2.5% paraformaldehyde/0.2% glutaraldehyde in 0.1 M phosphate buffer (pH 7). Pellets were embedded in HM20, 70-nm sections cut and examined using a Philips CM10 transmission electron microscope equipped with a Gatan Bioscan Camera.

## Endothelial cells from patients

The study was performed with the approval of the South-East Scotland Research Ethics Committee, in accordance with the Declaration of Helsinki and with the written informed consent of all participants. Patients with premature coronary artery disease and a family history of premature coronary artery disease ($n = 8$) were identified from the outpatient department, Royal Infirmary of Edinburgh, Scotland, UK. A control group of healthy age- and sex-matched subjects ($n = 8$) with no evidence of significant coronary artery disease following computed tomography coronary angiography (CTCA) was recruited from the Clinical Research Imaging Centre, Royal Infirmary of Edinburgh. Subjects attended the Clinical Research Facility at the Royal Infirmary of Edinburgh for vascular assessment and tissue sampling. Vessel wall endothelial cells were isolated from superficial forearm veins by wire biopsy for in vitro expansion as reported in Ref [48]. Briefly, under local anaesthetic, an 18-gauge venous cannula was inserted into a superficial forearm vein and a J-shaped guidewire passed and gently manipulated to harvest endothelial cells. EGM-2 medium was syringed through the wire to detach cells, which were collected by centrifugation and seeded into BD BioCoat Collagen 1 coated six-well plates (BD Biosciences). Cells were incubated under standard culture conditions in EGM-2 medium.

## Non-radioactive long-lived protein degradation assay

A non-radioactive pulse-chase protocol using L-azidohomoalanine (AHA) labelling was performed to quantify long-lived protein degradation during autophagy [32]. Cells were cultured in L-methionine-free medium for 30 min to deplete the intracellular methionine reserves. Following methionine depletion, the cells were labelled with AHA 18 h. Dialysed FBS (Thermo Fisher) was used to eliminate L-methionine from this other source. After labelling, the cells were cultured in regular medium or HBSS containing 10× L-methionine (2 mM) for 4 h to chase out the short-lived proteins. Then, cells were fixed in 4% formaldehyde in PBS and undergo a "click" reaction between the azide group of AHA and a fluorescently tagged alkyne probe (Click-iT® AHA; Thermo Fisher, C10102). The degradation of AHA-containing proteins was then detected by flow cytometry.

## Autophagic flux analysis by quantitative ratiometric flow cytometry

Cells stably expressing mCherry-GFP-LC3 were used for flow cytometric analysis. Briefly, a BD LSR Fortessa (Beckman Coulter) using 488 and 561 nm lasers for green and red fluorophore excitation,

respectively, was used to perform flow cytometry. The appropriate forward/side scatter profile was used to exclude non-viable cells. Cells undergoing autophagy were defined as those expressing a high mCherry/GFP fluorescence ratio, since delivery by autophagy to the lysosome quenches the GFP signal, but not the mCherry signal. The gates to define what constituted an increased mCherry/GFP fluorescence ratio were set based on cells treated with BafA1. The bottom of the gate for each set of flow cytometry experiments was therefore set at the rightward base of the BafA1-treated curve.

### TCHP knock-out mice tissues and histological analysis

Heart, liver and pancreas tissues are from C57BL/6N-Atm1Brd Tchp$^{tm1b\ (EUCOMM)Wtsi/WtsiIeg}$ (Tchp knock-out) mice and were kindly provided from the Wellcome Trust Sanger Institute. Histological analysis was performed on four wild-type mice (two male and two female) and four Tchp knock-out mice (two male and two female) 16-week-old.

Paraffin cross-sections were blocked with normal goat serum, incubated with anti-CD31 (Abcam; 1:200) and anti-p62 (Genetex 1:200) primary antibody overnight at 4°C, and then incubated with Alexa 488-conjugated anti-rat IgG antibody (Thermo Fisher). High-power fields were captured (at 400×), and the number of capillaries and arterioles per field was counted. At least 30 randomly chosen fields were evaluated per sample, in a blinded experiment. CD31-positive vessel area was quantified using ImageJ software and expressed per square micrometres.

### Statistical analysis

Comparisons between different conditions were assessed using the 2-tailed Student's *t*-test. If the normality test failed, the Mann–Whitney test was performed. Continuous data are expressed as mean ± SD of at least three independent experiments. *P*-value < 0.05 was considered statistically significant. Analyses were performed using GraphPad Prism v5.01.

## Data availability

The mass spectrometry proteomics data from this publication have been deposited to the ProteomeXchange Consortium (http://proteomecentral.proteomexchange.org) via the PRIDE partner repository [51] with the dataset identifier PXD015581.

**Expanded View** for this article is available online.

## Acknowledgements

We acknowledge the assistance of the microscopy and histology core facility at the University of Aberdeen, Lorraine Rose for technical help and Dr Shonna Johnston of the QMRI Flow Cytometry and Sorting Facility at the University of Edinburgh. This study was supported by grants from British Heart Foundation (BHF) (FS/11/52/29018) for A.C. and (FS/14/7/30574) for A.M; A.C. and N.O.C. acknowledge the support of the Wellcome Trust-University of Edinburgh Institutional Strategic Support Fund (ISSF3) and MRC-IMPC Pump Priming Award (MR/R014353/1). T.M is supported by BHF Career Re-entry Fellow (FS/16/38/32351); M.B. is a BHF Intermediate Fellow (FS/16/4/31831). A.C. acknowledges Edinburgh BHF Research Excellence Award.

## Author contributions

AM acquired and analysed the data and drafted the manuscript. AL; DDA; MV; NG, TM; DM; EP; MB; NLM; and JCD participated in acquiring and analyses of the data and revised the manuscript. NOC and JCD designed the phenotypic screening assays and high content data analysis protocols and contributed to revising the manuscript. LI performed the mass spectrometry experiments. AC designed the overall study and drafted the manuscript.

## Conflict of interest

The authors declare that they have no conflict of interest.

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
