## [Review Process File · EMBO Reports]

Trichoplein binds PCM1 and controls endothelial cell function by regulating autophagy

Andrea Martello, Angela Lauriola, David Mellis, Elisa Parish, John C Dawson, Lisa Imrie, Martina Vidmar, Noor Gammoh, Tijana Mitic, Mairi Brittan, Nicholas L Mills, Neil O Carragher, Domenico D'Arca, Andrea Caporali

Review timeline:

Submission date:	29 March 2019
Editorial Decision:	21 June 2019
Revision received:	2 October 2019
Editorial Decision:	20 December 2019
Revision received:	18 February 2020
Additional Correspondence	19 March 2020
Additional Correspondence	25 March 2020
Accepted:	31 March 2020

Transaction Report:

1st Editorial Decision

21 June 2019

Thank you for the submission of your research manuscript to our journal. I apologize for the unusual delay in handling your manuscript. It has been sent to three referees but despite many reminders referee 3, an expert in endothelial cell function, has not yet delivered his/her report. For the sake of time, I have thus decided to make a decision based on the two reports we have received (copied below). Please note that this is a preliminary decision made in the interest of time, and that it is subject to change should the third referee offer very strong and convincing reasons for this. As soon as we will receive the final report on your manuscript, we will forward it to you as well.

As you will see, both referees acknowledge that the findings are potentially interesting. However, they also point out a number of concerns and have a number of suggestions on how to strengthen and substantiate the work. Referee 1 is concerned that the effect of TCHP on autophagy is an indirect consequence of impaired microtubule-dependent transport of autophagosomes and lysosomes and that the current dataset is not sufficient to support a direct role for TCHP in autophagy. Referee 1 also points out that the overexpression of p62 can cause artefacts and that endogenous p62 should be used. Both referees emphasise that the analysis of autophagy requires further assays and thorough quantification. Moreover, both referees are concerned about the quality of the Western blots and the specificity of the PCM1 bands.

From the referee comments it is clear that a major revision will be required to substantiate the findings and that publication cannot be considered at this stage. Yet, given the potential interest of your findings and the constructive comments from the referees, I would like to give you the opportunity to address the concerns and would be willing to consider a revised manuscript with the understanding that the referee concerns must be fully addressed and their suggestions (as detailed above and in their reports) taken on board.

Should you decide to embark on such a revision, acceptance of the manuscript will depend on a

positive outcome of a second round of review and I should also remind you that it is EMBO reports policy to allow a single round of revision only and that, therefore, acceptance or rejection of the manuscript will depend on the completeness of your responses included in the next, final version of the manuscript.

Revised manuscripts should be submitted within three months of a request for revision; they will otherwise be treated as new submissions. Please contact us if a 3-months time frame is not sufficient for the revisions so that we can discuss the revisions further.

- 1) a .docx formatted version of the manuscript text (including legends for main figures, EV figures and tables). Please make sure that the changes are highlighted to be clearly visible.
- 2) individual production quality figure files as .eps, .tif, .jpg (one file per figure).
- 3) a .docx formatted letter INCLUDING the reviewers' reports and your detailed point-by-point responses to their comments. As part of the EMBO Press transparent editorial process, the point-by-point response is part of the Review Process File (RPF), which will be published alongside your paper.
- 4) a complete author checklist, which you can download from our author guidelines (<<http://embor.embopress.org/authorguide>>). Please insert information in the checklist that is also reflected in the manuscript. The completed author checklist will also be part of the RPF.
- 5) Please note that all corresponding authors are required to supply an ORCID ID for their name upon submission of a revised manuscript (<<https://orcid.org/>>). Please find instructions on how to link your ORCID ID to your account in our manuscript tracking system in our Author guidelines (<<http://embor.embopress.org/authorguide>>).
- 6) We replaced Supplementary Information with Expanded View (EV) Figures and Tables that are collapsible/expandable online. A maximum of 5 EV Figures can be typeset. EV Figures should be cited as 'Figure EV1, Figure EV2' etc... in the text and their respective legends should be included in the main text after the legends of regular figures.

- For the figures that you do NOT wish to display as Expanded View figures, they should be bundled together with their legends in a single PDF file called *Appendix*, which should start with a short Table of Content. Appendix figures should be referred to in the main text as: "Appendix Figure S1, Appendix Figure S2" etc. See detailed instructions regarding expanded view here: <<http://embor.embopress.org/authorguide#expandedview>>.

- 7) Before submitting your revision, primary datasets (and computer code, where appropriate) produced in this study need to be deposited in an appropriate public database (see <<http://embor.embopress.org/authorguide#dataavailability>>).

Specifically, we would kindly ask you to provide public access to the proteomics dataset.

The accession numbers and database should be listed in a formal "Data Availability" section (placed after Materials & Method) that follows the model below (see also <<http://msb.embopress.org/authorguide#dataavailability>>). Please note that the Data Availability Section is restricted to new primary data that are part of this study.

Data availability

8) We would also encourage you to include the source data for figure panels that show essential data. Numerical data should be provided as individual .xls or .csv files (including a tab describing the data). For blots or microscopy, uncropped images should be submitted (using a zip archive if multiple images need to be supplied for one panel). Additional information on source data and instruction on how to label the files are available <http://embor.embopress.org/authorguide#sourcedata>.

9)) Regarding data quantification:

- Please ensure to specify the name of the statistical test used to generate error bars and P values, the number (n) of independent experiments underlying each data point (not replicate measures of one sample), and the test used to calculate p-values in each figure legend. Discussion of statistical methodology can be reported in the materials and methods section, but figure legends should contain a basic description of n, P and the test applied.
- IMPORTANT: Please note that error bars and statistical comparisons may only be applied to data obtained from at least three independent biological replicates. If the data rely on a smaller number of replicates, scatter blots showing individual data points are recommended.
- Graphs must include a description of the bars and the error bars (s.d., s.e.m.).
- Please also include scale bars in all microscopy images.

10) As part of the EMBO publication's Transparent Editorial Process, EMBO reports publishes online a Review Process File to accompany accepted manuscripts. This File will be published in conjunction with your paper and will include the referee reports, your point-by-point response and all pertinent correspondence relating to the manuscript.

I look forward to seeing a revised version of your manuscript when it is ready. Please let me know if you have questions or comments regarding the revision.

REFeree REPORTS

Referee #1:

Summary:

Martello et al describe a role of Trichoplein (TCHP) linking autophagy and vascular function in endothelial cells (ECs). The effects the authors observe are a defect in autophagosome maturation when TCHP is depleted and destabilization of PCM1 and subsequent destabilization of GABARAP. This results in a blockage of the latter steps of autophagosome maturation, the fusion with the lysosome. Interestingly the autophagy defects occur only under nutrient rich conditions, which can be overcome by starvation treatment.

Comments to the authors:

This is a nice paper with plenty of good data that somewhat backs up their main point. I especially like the inclusion of the patient data as this lends strength to the role of TCHP in regulating p62 stability.

However, I find the declaration of "TCHP is indispensable for autophagosome maturation" a bit on the strong side especially considering that the defect can be overcome and normal autophagy processing of p62/LC3B by starvation treatment. If it was the case that TCHP was indispensable then starvation would not have been able to overcome the maturation defect.

I also find that the direct link between TCHP and autophagy is a little misleading as it seems to be a more indirect effect. TCHP controls PCM1 which controls GABARAP stability. Is the function of TCHP really just to control PCM1 localization and the effect on GABARAP/autophagy is really just a secondary effect?

I think the mechanisms of TCHP function could be explored which may lead to the real reason for the defect in autophagy. For example, the destabilization of PCM1, which localizes to the microtubule organizing centre, may actually affect the microtubule network. This then affects both autophagosome transport and lysosome positioning, both of which can actually explain the defect they authors observe.

In figure S2B there is clear perinuclear accumulation of both Rab7 and LAMP2 (lysosomes) which is indicative of inhibition of MT plus end transport. Lysosome positioning is essential for its function and also its degradative capacity and fusion with both endocytic and autophagy compartments. The authors should use FCS only starvation and the feeding assay and look at the position of LAMP2 vesicles. This will tell you whether lysosome trafficking is affected. The authors would also need to look at tubulin staining for the MT network.

Figure comments.

Figure 2: A - The IP sample of TCHP (PCM1 blot) there are multiple bands/smears. Which one is the relevant band? Are there lower exposures?

(B) Again the PCM1 blot looks like a large smear across two lanes. This makes it difficult to interpret etc. Lower exposure?

Figure 4 - (C) There are now robust protocols for the analysis of tandem-tagged LC3b by flow cytometry. Could the authors use this to quantify instead of manual counting procedures?

(D) - The authors have overexpressed p62. This can be problematic due to aggregation of p62 due to both its N-terminal PB1 and c-terminal UBA domain. The authors should use endogenous p62 and not overexpressed.

Figure 5:C - the individual channels should be shown in grey scale and if possible, merged should be in colour blind friendly colours (CMYK). Also, in (i) there looks to be almost 100% colocalization between p62 and lc3B, this is what I would expect as p62 aggregates and attracts LC3B membranes. It is highly unusual to see lc3b negative p62 puncta and I am not sure of the relevance of this. Did the authors also look at ubiquitin as well?

The authors should consider using Syntaxin17 as marker for complete autophagosomes that haven't yet fused with lysosomes. This could be indicative of the impaired fusion step.

Figure 6. This is good data, however, it does not seem to fit well with the narrative of the paper. I feel that the drug screen is perhaps part of a larger story that would be better suited as a manuscript on its own as it requires a lot more validation etc. This just led to a slightly confused and unclear message towards the end of the paper

Figure S3 is labelled incorrectly in the figure legends.

Other comments:

The language could use a bit of tidying up throughout. Perhaps the use of some online tools (such as Grammarly) could better assist the authors in this regard?

Overall, I like a lot of the work that the authors have put together, however I feel that it is more an indirect effect of TCHP1 on autophagy rather than being a direct regulator of the process. There are a few issues, such as the mechanisms of TCHP1 function, that require to be addressed prior to acceptance. I also feel the focus of the paper is lost a little bit, particularly for a scientific report in EMBO reports.

I think there would be sufficient broad interest in the paper, due its role in autophagy (?), and potentially vesicle trafficking, and after revision may be more acceptable for publication.

Referee #2:

Martello et al. characterized the role of trichoplein (TCHP) in endothelial cells and demonstrate that capillary-like structures are less efficiently formed and migration ability is reduced upon TCHP knockdown (figure 1). In order to further address the functional role of TCHP in endothelial cells the group conducted a proteomics analysis and identified PCM1 as a candidate protein-protein interaction partner of TCHP (Table 1, figure 2). As it has been already shown that PCM1 is implicated in autophagy, the authors addressed the role of TCHP in autophagy and in the endolysosomal compartment (figure 4, 5). Based on this assessment (GABARAP, LC3B, p62 western blotting and intracellular localization analysis) Martello et al. suggest that TCHP should be involved in the control of basal autophagy as p62 and LC3B levels are increased in basal autophagy conditions. In line with this observation p62 was increased in TCHP-deficient mice (heart and cardiac vessels) and in coronary artery disease patient-derived endothelial cells (figure 7). The group also screened for compounds that restore normal p62 level in the absence of TCHP and identified BAY11-7082, an NFκB inhibitor. Moreover, conducting CHIP-qPCR the authors observed an enrichment of NFκB at the p62 promotor region in the absence of TCHP (figure 6). Overall, the suggested outcome of the study is very interesting but the authors should thoroughly revise their report as several experiments appear not convincing at this stage.

Major points

Table 1

- Display of proteomics results should be detailed, extended and raw data should be uploaded at appropriate platforms. The complete list of candidate protein-protein interaction partners of TCHP should be shown and the putative protein network appropriately displayed e.g. in a separate figure.
- Figure 2
- PCM1 protein bands in 2A appear obscure. Several bands are present in HEK293 cells but not in HUVECs. The authors should clearly label specific PCM1 bands in 2A, but also in 2B. Also, marker band positions are missing in 2C.
 - The authors used truncated TCHP variants in 2B. At least by homology modelling the authors should address if these variants may fold properly.
 - Colocalizations in 2D and 2E require a thorough quantification, in particular because several intracellular locations have been suggested for TCHP (keratin, microtubules, centrioles, ER/mitochondria, desmosomes) and it needs to be clarified where TCHP and PSM1 colocalize.
- Figure 3
- GABARAP protein bands appear obscure, the authors should repeat these experiments and clearly show and quantify GABARAP-I and GABARAP-II bands (or is the band displayed pro-GABARAP?). More proteasome inhibitors should be used, and GABARAP mRNA levels quantified in parallel.
- Figure 4
- As the authors suggest that basal autophagy is compromised in the absence of TCHP more autophagy marker/assays should be employed to clearly demonstrate this point.
- Figure 5
- Quality of autophagosome fraction should be addressed e.g. images (EM) derived from these fractions should be shown and/or at least autophagy marker proteins used for western blotting (LC3/GABARAP).
- Figure 6
- NFκB should be downregulated and p62 expression analyzed.

Minor points

- Title and abstract read very broad and should be adjusted to more accurately summarize the main findings.
- Method section need to be thoroughly extended to provide more details.
- The new study by Wirth et al (Nat Commun 2019) on LC3/GABARAP members and PCM1 should be included and discussed.

Referee #1:**Summary:**

Martello et al describe a role of Trichoplein (TCHP) linking autophagy and vascular function in endothelial cells (ECs). The effects the authors observe are a defect in autophagosome maturation when TCHP is depleted and destabilization of PCMI and subsequent destabilization of GABARAP. This results in a blockage of the latter steps of autophagosome maturation, the fusion with the lysosome. Interestingly the autophagy defects occur only under nutrient rich conditions, which can be overcome by starvation treatment.

Comments to the authors:

This is a nice paper with plenty of good data that somewhat backs up their main point. I especially like the inclusion of the patient data as this lends strength to the role of TCHP in regulating p62 stability.

We thank this reviewer for the positive comments and suggestions which have helped us to improve our manuscript.

However, I find the declaration of "TCHP is indispensable for autophagosome maturation" a bit on the strong side especially considering that the defect can be overcome and normal autophagy processing of p62/LC3B by starvation treatment. If it was the case that TCHP was indispensable, then starvation would not have been able to overcome the maturation defect. I also find that the direct link between TCHP and autophagy is a little misleading as it seems to be a more indirect effect. TCHP controls PCMI which controls GABARAP stability. Is the function of TCHP really just to control PCMI localization and the effect on GABARAP/autophagy is really just a secondary effect?

We thank the reviewer for discussing this point. We agreed that TCHP has an indirect role in controlling autophagy. Therefore, we have amended the text and corrected abstract accordingly. One of the key points of this study is the localization of TCHP in the centriolar satellites together with PCMI. We observed that TCHP determines PCMI-centriolar satellite localization and its stability since PCMI is lost from the satellites and is degraded when TCHP is depleted. Collectively, these sets of events lead to the destabilization of GABARAP and consequently to defective autophagy in TCHP knock-down cells. Moreover, we have also reinforced the contribution of TCHP to autophagy pathway by including new autophagy assays and improving the analysis of autophagy phenotype in TCHP knock-down cells.

I think the mechanisms of TCHP function could be explored which may lead to the real reason for the defect in autophagy. For example, the destabilization of PCMI, which localizes to the microtubule organizing centre, may actually affect the microtubule network. This then affects both autophagosome transport and lysosome positioning, both of which can actually explain the defect they authors observe. In figure S2B there is clear perinuclear accumulation of both Rab7 and LAMP2 (lysosomes) which is indicative of inhibition of MT plus end transport. Lysosome positioning is essential for its function and also its degradative capacity and fusion with both endocytic and autophagy compartments. The authors should use FCS only starvation and the feeding assay and look at the position of LAMP2 vesicles. This will tell you whether lysosome trafficking is affected. The authors would also need to look at tubulin staining for the MT network.

As suggested by the reviewer, we agreed that the proposed experiments are important to clarify the phenotype of the TCHP knock-down cells. In particular, it has previously been reported that TCHP controls the positioning and anchoring of MT at the centrosome (Ibi M et al. JCS 2011). In line with this, we observed an impaired MT network in growing conditions of TCHP knock-down cells which is not modulated by serum starvation (Figure EV4D).

Analysis of lysosomes distribution showed that, in control cells, the starvation increased the proportion of cells with predominantly perinuclear lysosomes, while upon recovery, LAMP2-positive vesicles localized at the cell periphery (Fig EV4D). On the contrary, TCHP knock-down cells showed that the perinuclear distribution of lysosome in growing condition (full medium) which is not changing during starvation or recovery conditions (Fig EV4D).

Despite TCHP depleted cells are showing widespread alteration of lysosomal distribution; we still observed a strong activation of the endo-lysosomal pathway, lysosomal activity and acidification (Figure EV4B). The lysosomal activity was further validated by an observed increase in the degradation of EGFR in the TCHP knock-down cells (Figure EV4C). Therefore, our findings are ruling out lysosomal dysfunction as the cause of decreased autophagic flux. Finally, p62 level in TCHP knock-down cells decreased in serum starvation conditions as in control cells (Fig EV4E), showing that in TCHP knock-down cells the autophagic process is not affected by the limited mobility of lysosomes and the defective MT network.

Figure comments.

Figure 2: A - The IP sample of TCHP (PCM1 blot) there are multiple bands/smears. Which one is the relevant band? Are there lower exposures? (B) Again the PCM1 blot looks like a large smear across two lanes. This makes it difficult to interpret etc. Lower exposure?

We are now showing in Figure 1A, and 1B the immunoblots for PCM1 after TCHP-FLAG pull down with a shorter exposure. We detected a single specific band for PCM1 at 230 KDa. (Cell Signalling Technology antibody, PCM1 – G2000; cat# 5213).

Figure 4 - (C) There are now robust protocols for the analysis of tandem-tagged LC3b by flow cytometry. Could the authors use this to quantify instead of manual counting procedures?

We thank the reviewer for the suggestion and in addition to the analysis using confocal microscopy we have now included the experiments with the mCherry-GFP-LC3 tandem reporter, using quantitative ratiometric analysis by flow cytometric (Figure 3D) (protocol from Gump JM et al. Autophagy 2014 added to the Methods section). The mCherry/GFP fluorescence ratio was lower in the TCHP knock-down cells compared with the control cells, attesting to a decrease in autolysosome formation, whereas the ratio increased in both TCHP knock-down and control cells following starvation in HBSS medium, showing that the block of autophagic flux is reversed by starvation.

(D) - The authors have overexpressed p62. This can be problematic due to aggregation of p62 due to both its N-terminal PB1 and c-terminal UBA domain. The authors should use endogenous p62 and not overexpressed.

As suggested, we have removed the experiment with exogenous p62. The analysis of the levels of endogenous p62 during modulation of autophagy was already presented in Figure 4B (now Figure 3B) in the first version of the manuscript. In a complete medium, Western blot analysis confirmed an increased level of p62 protein in TCHP knock-down cells compared with control. When autophagic flux was blocked with bafilomycin A1 at basal conditions, there was a greater accumulation of p62 in the control cells compared with that in TCHP knock-down cells, while this difference was blunted after induction of autophagy (Figure 3B).

Figure 5:C - the individual channels should be shown in grey scale and if possible, merged should be in colour blind friendly colours (CMYK). Also, in (i) there looks to be almost 100% colocalization between p62 and lc3B, this is what I would expect as p62 aggregates and attracts LC3B membranes. It is highly unusual to see lc3b negative p62 puncta and I am not sure of the relevance of this. Did the authors also look at ubiquitin as well?

Images in Figure 5 (now Figure EV3A-C) have now been updated with individual channels showing in greyscale and merged channels in CMYK (i.e. colour-blind friendly colours). The same setting has been applied to all the images throughout the paper. Additionally, we have included images of co-localization of ubiquitin and p62 puncta in TCHP knock-down and control cells instead of LC3 and p62 co-localization. Results are reported in Figure EV3C, and we observed a reduced co-localization between p62 and ubiquitin puncta in cells lacking TCHP.

The authors should consider using Syntaxin17 as marker for complete autophagosomes that haven't yet fused with lysosomes. This could be indicative of the impaired fusion step.

As suggested, we have analysed the co-localization of LC3 and Syntaxin 17 (STX17) in TCHP knock-down cells. Cells were transduced with GFP-LC3 retrovirus and immunostained for STX17 (Figure 4C). We observed a significant decrease in the number of cells with double-positive for LC3 and STX17 in TCHP knock-down cells at the basal condition and after BafA1 treatment (Fig 4C). Moreover, STX17 recruitment to the autophagosome (LC3/STX17 positive puncta per cells) is disrupted in TCHP knock-down cells during the BafA1 treatment (Fig 4C), therefore showing that the formation of mature autophagosomes is inhibited in TCHP knock-down cells.

Figure 6. This is good data, however, it does not seem to fit well with the narrative of the paper. I feel that the drug screen is perhaps part of a larger story that would better suited as a manuscript on its own as it requires a lot more validation etc. This just led to a slightly confused and unclear message towards the end of the paper.

We thank the reviewer for raising this point. We have moved the Figure 6 in Expanded View section as Figure EV5. We believe that these experiments are still critical to explain the phenotype in endothelial cells from patients with coronary artery disease and therefore, we have kept them in the final version of the manuscript.

Figure S3 is labelled incorrectly in the figure legends.

Figure S3 has now been moved to Appendix Section (see Appendix Figure S4) and figure legend has been amended accordingly.

Other comments:

The language could use a bit of tidying up throughout. Perhaps the use of some online tools (such as Grammarly) could better assist the authors in this regard?

Language revision has been performed throughout the text with the support of English speakers and Grammarly online tool.

Overall, I like a lot of the work that the authors have put together, however I feel that it is more an indirect effect of TCHP1 on autophagy rather than being a direct regulator of the process. There are a few issues, such as the mechanisms of TCHP1 function, that require to be addressed prior to acceptance. I also feel the focus of the paper is lost a little bit, particularly for a scientific report in EMBO reports. I think there would be sufficient broad interest in the paper, due its role in autophagy (?), and potentially vesicle trafficking, and after revision may be more acceptable for publication.

We thank this reviewer for understanding our manuscript and for their positive comments. We have found their suggestions insightful, and we believe we were able to put our observations in a better context and have improved the manuscript accordingly. The manuscript has been edited to reach the EMBO Reports readers.

Referee #2:

Martello et al. characterized the role of trichoplein (TCHP) in endothelial cells and demonstrate that capillary-like structures are less efficiently formed and migration ability is reduced upon TCHP knockdown (figure 1). In order to further address the functional role of TCHP in endothelial cells the group conducted a proteomics analysis and identified PCMI as a candidate protein-protein interaction partner of TCHP (Table 1, figure 2). As it has been already shown that PCMI is implicated in autophagy, the authors addressed the role of TCHP in autophagy and in the endolysosomal compartment (figure 4, 5). Based on this assessment (GABARAP, LC3B, p62 western blotting and intracellular localization analysis) Martello et al. suggest that TCHP should be involved in the control of basal autophagy as p62 and LC3B levels are increased in basal autophagy conditions. In line with this observation p62 was increased in TCHP-deficient mice (heart and cardiac vessels) and in coronary artery disease patient-derived endothelial cells (figure 7). The group also screened for compounds that restore normal p62 level in the absence of TCHP and identified BAY11-7082, an NFkB inhibitor. Moreover, conducting CHIP-qPCR the authors observed an enrichment of NFkB at the p62 promotor region in the absence of TCHP (figure 6). Overall, the suggested outcome of the study is very interesting, but the authors should thoroughly revise their report as several experiments appear not convincing at this stage.

We thank the reviewer for positive comments on our study. We believe we have now addressed all the points raised in their comments and have improved the manuscript accordingly.

Major points

Table 1

Display of proteomics results should be detailed, extended and raw data should be uploaded at appropriate platforms. The complete list of candidate protein-protein interaction partners of

TCHP should be shown and the putative protein network appropriately displayed e.g. in a separate figure.

As requested, the mass spectrometry proteomics data have been deposited to the ProteomeXchange Consortium via the PRIDE partner repository with the dataset identifier PXD015581 (Reviewer account details have been provided with the cover letter). We have now expanded Table EV1 showing the complete list of TCHP protein partners. Moreover, the analysis of the functional protein association network has been performed, and network image generated using STRING (<https://string-db.org/>), shown in Appendix Figure S1A.

Figure 2

• PCM1 protein bands in 2A appear obscure. Several bands are present in HEK293 cells but not in HUVECs. The authors should clearly label specific PCM1 bands in 2A, but also in 2B. Also, marker band positions are missing in 2C.

The updated Figures 1A and 1B are now showing immunoblots for PCM1 after TCHP-FLAG pull down with a shorter exposure. We detected a single specific band for PCM1 at 230 KDa (Cell Signalling Technology antibody (PCM1 – G2000; #5213). Molecular weight has been added in Figure 1C.

• The authors used truncated TCHP variants in 2B. At least by homology modelling the authors should address if these variants may fold properly.

The SWISS-MODEL (<https://swissmodel.expasy.org/>) template library (SMTL version 18-07-2019, PDB release 12-7-2019) was searched with BLAST (Camacho, C et al. 2009) and HHBlits (Remmert et al. 2012) for related evolutionary structures matching the target sequence for TCHP Δ 1 and TCHP Δ 1,2. The top hits homologous proteins were Survivin for TCHP Δ 1 and ATP synthase subunit b for TCHP Δ 1,2. Model figures have been included in Appendix Figure S1B.

• Colocalizations in 2D and 2E require a thorough quantification, in particular because several intracellular locations have been suggested for TCHP (keratin, microtubules, centrioles, ER/mitochondria, desmosomes) and it needs to be clarified where TCHP and PSM1 colocalize.

PCM1 was the first identified satellite protein and is acting as a master satellite assembly scaffold (Balczon et al, 1994), since it interacts with and is required for the localization of a number of other satellite components, including CEP290 (Kim et al, 2008) and CEP72 (Stowe et al, 2012). We demonstrated that TCHP extensively co-localized with PCM1 in the pericentriolar matrix and satellite region (Fig 1D) and co-localized also with centriolar satellite proteins CEP290 and CEP72 (Fig 1D), therefore showing for the first time TCHP localization at the centriolar satellites. During the revision of this manuscript, an independent study was published reporting the spatial and proteomic profiling of 22 human satellite proteins by proximity-dependent biotin identification (BioID), including PCM1, CEP290 and CEP72 (Gheiratmand et al, 2019). Consistently with our results, the aforementioned unbiased approach has identified TCHP as part of the centriolar satellite protein network, directly interacting with PCM1, CEP290 and CEP72.

Quantification of the co-localisation between TCHP and PCM1, CEP72 and CEP290 dots has been now added in Figure 1D. We have further provided the quantification graph associated with Figure 1E.

Figure 3

• GABARAP protein bands appear obscure, the authors should repeat these experiments and clearly show and quantify GABARAP-I and GABARAP-II bands (or is the band displayed pro-GABARAP?). More proteasome inhibitors should be used, and GABARAP mRNA levels quantified in parallel.

Experiments comprising Western blots analysis of GABARAP stability have been repeated and new blots are displayed in Figure 2B and 2D. MG132 proteasomal inhibitor has been used at 2 μ M concentration for the experiments presented in Figure 2. This concentration was sufficient to inhibit proteasomal degradation without inducing toxicity in primary endothelial cells. The overall quality of the Western blot bands has improved; however, despite the longer exposure, we were not able to detect the doublet bands representing GABARAP-I and -II in endothelial cells using the GABARAP Cell Signalling Technology antibody (E1J4E, cat #13733) (Figure 2B and 2D). GABARAP-II band appeared only when the endothelial cells are treated with BafA1 in low speed and high speed fractions as showed in Figure 5A (Protection assay). Therefore, we can assume that the band that is normally detected in endothelial cells is GABARAP-I.

Finally, the relative expression of PCMI and GABARAP mRNA levels have been shown in Appendix Figure S1A and S1B, respectively.

Figure 4

• **As the authors suggest that basal autophagy is compromised in the absence of TCHP more autophagy marker/assays should be employed to clearly demonstrate this point.**

One of the gold standard methods for measuring autophagic flux involves a direct measurement and quantification of increased proteolysis for long-lived proteins (Wang J et al. Nature Prot 2017). Hence, to confirm the role of TCHP in regulating autophagic flux, we performed a nonradioactive pulse-chase protocol using L-azidohomoalanine (AHA) labelling (Figure 3E and Figure EV2B). AHA is used as a surrogate for L-methionine, and, when added to the cultured cells grown in methionine-free medium, AHA gets incorporated into proteins during *de novo* protein synthesis. After a chase period to remove short-lived proteins, autophagy was induced by HBSS treatment. Cells then undergo a 'click' reaction between the azide group of AHA and a fluorescently tagged alkyne probe. The AHA-containing proteins were then analysed by flow cytometry, quantification seen in Figure 3E. TCHP knock-down cells with AHA labelling that have undergone a click reaction had a substantial increase in accumulated protein within the cells (seen through increase cellular fluorescence intensity, Figure 3E and Figure EV2B) as compared with the control sample, whereas the cells that underwent amino acid starvation for 4 hours had reduced accumulation of protein (reduced fluorescence intensity).

Figure 5

• **Quality of autophagosome fraction should be addressed e.g. images (EM) derived from these fractions should be shown and/or at least autophagy marker proteins used for western blotting (LC3/GABARAP).**

We agreed with this comment, and we have now provided, together with p62, the Western blots for LC3 and GABARAP detection in low- (LSP) and high- speed pellets (HSP) of TCHP knock-down cells vs. control in the presence or absence of bafilomycin A1, see Figure 4A.

Figure 6

• **NFkB should be downregulated and p62 expression analyzed.**

As suggested by the Reviewer, we have silenced NF-KBp65 using siRNA approach, and we subsequently measured p62 expression in the TCHP knock-down and control cells. The silencing of NF-KBp65 reduced the expression of p62 in TCHP knock-down cells (Figure EV5F).

Minor points

• **Title and abstract read very broad and should be adjusted to more accurately summarize the main findings.**

The title has been changed to: "*Trichoplein binds PCMI and controls endothelial cell function by regulating autophagy*". The abstract has also been modified to include the main findings of the study.

• **Method section need to be thoroughly extended to provide more details.**

Methods section has been expanded to include mass spectrometry and imaging analysis protocols. All the new experiments have been included in the Materials & Methods section.

• **The new study by Wirth et al (Nat Commun 2019) on LC3/GABARAP members and PCMI should be included and discussed.**

The paper by Wirth et al. has been included and discussed on page 12 in the Discussion section.

Thank you for the submission of your revised manuscript to EMBO reports. I apologize for the unusual delay in handling your manuscript but we have only now received the second referee report. Please find both reports copied below my signature.

As you will see, both referees are positive about the study and support publication after some further clarification of text and figures. Referee 1 suggests testing whether reconstitution of the TCHP-depleted cells with WT or mutant THCP rescues PCMI levels and autophagy, which appears straightforward and should be added.

As per our editorial policies we allow to use bar graphs for data based on $n=3$, but the plotting of individual data points as suggested by referee 1 would certainly add value and transparency to the data.

From the editorial side, there are also a few things that we need before we can proceed with the official acceptance of your study.

REFEREE REPORTS

Referee #1:

The revised manuscript by Martello et al., has addressed my initial concerns and I am happy that they have adjusted their title to more accurately reflect the data presented. The premise of the paper is that the interaction of TCHP with PCMI indirectly regulates basal autophagy through controlling GABARAP stability. Overall, the data regarding the impact of TCHP loss on autophagy is now convincing and the inclusion of the patient data is an excellent inclusion. Perhaps where the paper could be stronger is the link between TCHP and PCMI. The authors clearly show that the two proteins interact and mapped the interaction site on TCHP to the second coil-coil domain. However, the authors have not shown that reconstitution of the TCHP depleted cells with WT or mutant (dCC1, dCC2) TCHP restores (or not) PCMI localization and basal autophagy. This is perhaps the only missing part of the mechanism they are proposing and it would strengthen the paper.

Minor:

The format is not correct for a short report. The discussion should not be separate (Results and discussion sections should be combined).

Some of the IF images (e.g. Figure 3A, 3C) are not colour blind friendly and should be reformatted to have a consistent style throughout the manuscript (see figure 1D for example).

The authors change between displaying mean \pm SEM and Mean \pm SD in the same figure (Figure 3 and Figure 5), with no explanation as to why those have been chosen. Exact p values should be shown and also SEM/SD information is missing from a number of the figure sections (e.g. Figure 3B, 3C)

The authors have chosen bar graphs for the majority of $n=3$ studies, individual points should be plotted instead.

There is no statistical analysis section in the materials and methods section.

Referee #2:

The authors provide a thoroughly revised manuscript in both content and structure and the reviewer wishes to congratulate the authors for their interesting study.

As previously requested, the authors have

- Deposited their proteomics data (PRIDE: PXD01558, show the complete list of TCHP protein partners, and display their data using the STRING database.
- Improved the labelling of their western blots
- Provide SWISS-MODEL based homology modelling of TCHP variants used in their study
- Quantification of TCHP and PCMI, CEP72 and CEP290 puncta that colocalize
- GABARAP lipidation assessments have been improved
- Conducted an appropriate autophagy flux assay (AHA labelling)
- Provide EM images of autophagosome fractions
- Conducted functional NFkB assessments with regard to p62 expression

The experiments presented in this revised version coherently support the authors interpretation and the manuscript is suitable for publication.

Referee #1:

The revised manuscript by Martello et al., has addressed my initial concerns and I am happy that they have adjusted their title to more accurately reflect the data presented. The premise of the paper is that the interaction of TCHP with PCM1 indirectly regulates basal autophagy through controlling GABARAP stability. Overall, the data regarding the impact of TCHP loss on autophagy is now convincing and the inclusion of the patient data is an excellent inclusion. Perhaps where the paper could be stronger is the link between TCHP and PCM1. The authors clearly show that the two proteins interact and mapped the interaction site on TCHP to the second coil-coil domain.

Thanks for the positive comments on our study.

However, the authors have not shown that reconstitution of the TCHP depleted cells with WT or mutant (dCC1, dCC2) TCHP restores (or not) PCM1 localisation and basal autophagy. This is perhaps the only missing part of the mechanism they are proposing and it would strengthen the paper.

We understand the point the reviewer is making. To answer to review's comment, we have performed the requested experiment in endothelial cells using the exogenous wild-type TCHP, cloned into a lentiviral vector. The data shows that in endothelial cells lacking TCHP, overexpression of exogenous TCHP (WT) partially restores localisation of PCM1 in the centriolar satellites (new Fig EV2C) and decreases p62 accumulation (new Fig EV2D).

In the interest of time, we have tested the functionality of each mutant using HeLa cells before sub-cloning them in lentiviral vector. Indeed, lentiviral vector is the only efficient way to overexpress a gene in endothelial cells. Upon test experiments, we noticed that WT and TCHP Δ 1, which both bind PCM1, have accumulated in the PCM1-positive centriolar satellites. This is in line with our proposed mechanism. However, TCHP Δ 1,2 overexpression removed PCM1 from centriolar satellites (as observed in TCHP knock-down cells). Latter finding would suggest that the mutant is probably acting as a dominant-negative for TCHP or other centriolar satellite proteins. For this reason, we decided to not carry on with the experiments using plasmids carrying the TCHP mutant sequences in endothelial cells.

Minor:

The format is not correct for a short report. The discussion should not be separate (Results and discussion sections should be combined).

The format has been changed, and Results and Discussion sections are now combined.

Some of the IF images (e.g. Figure 3A, 3C) are not colour blind-friendly and should be reformatted to have a consistent style throughout the manuscript (see figure 1D for example).
The IF images have been modified in colour-blind friendly as suggested by this reviewer.

The authors change between displaying mean +/- SEM and Mean +/- SD in the same figure (Figure 3 and Figure 5), with no explanation as to why those have been chosen. Exact p values should be shown and also SEM/SD information is missing from a number of the figure sections (e.g Figure 3B, 3C).

We have now displayed the mean +/- SD for all the figures, and we added the exact p-value for each experiment.

The authors have chosen bar graphs for the majority of n=3 studies, individual points should be plotted instead.

The graphs have been modified, showing individual points for each experiment.

There is no statistical analysis section in the materials and methods section.

We have now added the statistical analysis method in the materials and methods section.

Additional correspondence

19 March 2020

I apologize again for the delay in handling your revised manuscript but I have discussed your manuscript again with referee 1 (see comments pasted below) and the editorial team.

We note that your data indicate that TCHP influences autophagy by stabilizing PCM1. You map the interaction between TCHP and PCM1 to the second coiled coil domain of TCHP in Figure 1B. The co-IP data indicate that TCHP lacking CC1 and CC2 cannot interact with PCM1 anymore in HEK cells. However, when tested in HeLa cells, this mutant version displaces PCM1 from centrioles. You indicated that this might be due to a dominant negative effect of TCHP dCC1, CC2.

However, we have to note that these experiments weaken (a) the causality, i.e., that TCHP acts via PCM1 and (b) the evidence that CC2 is indeed the domain that interacts with PCM1. TCHP dCC1, CC2 displaces PCM1 from centriolar satellites in HeLa cells but fails to interact with PCM1 in HEK cells.

Ideally the mutant protein should be tested in endothelial cells. At the least these issues should be discussed in the manuscript text and it might also be advisable to remove the interaction domain mapping.

I am happy to hear your thoughts on these issues and to discuss further with you.

Comments from referee 1:

I have a re-read through the paper with the new data and modifications. At this stage I am unsure about the supporting data and the mechanism they propose. For example, they were able to show that exogenous expression (using lentiviral vector TCHP WT) in TCHP restored the defects they seen. However, they tested the mutant forms of TCHP in HeLa cells that, presumably, still have endogenous TCHP. Therefore, they are really testing the effects of overexpressing the mutants and not the role of the individual mutants in the absence of TCHP. In my mind, they should have done this straight in the endothelial cells as this is the primary functional cell type they are testing.

This brings me back to the original point I made in my comment that the overall function of TCHP is to stabilise PCM1 and that the effects of TCHP loss on autophagy, although strongly supported, is a secondary effect and is dependent on on the stability of PCM1/GABARAP.

This would be absolutely fine IF they were able to show that the dCC2 form of TCHP (that doesn't interact with PCM1) is unable to rescue the effects on autophagy etc. This would nail the mechanism and give the good mechanistic insight into the effect TCHP loss was having and would therefore link very nicely to the clinical data.

I still don't understand why they did not express it in the endothelial cells lacking TCHP and if they seen the dom negative effect there that would also be more acceptable but would need clarification in the text.

Author response

25 March 2020

Thank you for the opportunity to discuss the issue in the text.

I agreed that it was better to leave Figure 1B and discuss the results in the manuscript. I have amended the manuscript accordingly and discussed the results on page 8 line 22.

Accepted

31 March 2020

I am very pleased to accept your manuscript for publication in the next available issue of EMBO reports. Thank you for your contribution to our journal.

Corresponding Author Name: Andrea Caporali

Manuscript Number: 2019-48192